# Anytime Safe PAC Efficient Reasoning

Chengyao Yu [1*]   Hao Zeng [1*]   Youxin Zhu [1]   Jianguo Huang [2]   Huajun Zeng [3]   Bingyi Jing [4 5]

## Abstract

Large Reasoning Models (LRMs) have demonstrated remarkable performance on complex tasks but suffer from high computational costs and latency. While selective thinking strategies improve efficiency by routing easy queries to non-thinking models, existing approaches often incur uncontrollable errors, especially in online settings where the performance loss of a non-thinking model is only partially observed and data are non-stationary. To address this, we propose *Betting Probably Approximately Correct* (B-PAC) *reasoning*, a principled method that enables anytime safe and efficient online reasoning under partial feedback. Specifically, we utilize inverse propensity scoring estimators to construct test supermartingales for candidate thresholds, and then dynamically adjust the routing threshold based on the accumulated statistical evidence of safety. Theoretically, we establish the anytime-valid performance loss control and the efficiency of B-PAC reasoning. Extensive experiments demonstrate that B-PAC reasoning significantly reduces computational overhead, decreasing thinking model usage by up to 81.01%, while controlling the performance loss below the user-specified level.

## 1. Introduction

Large reasoning models (LRMs) have exhibited remarkable capabilities on a range of challenging reasoning tasks, typically associated with generating long chain-of-thoughts (CoTs) (Guo et al., 2025; Yang et al., 2025; Jaech et al., 2024). Despite their powerful performance, LRMs tend to generate excessively long reasoning chains even for simple questions, referred to as the phenomenon of *overthinking* (Chen et al., 2024; Sui et al., 2025), resulting in substantial computational overhead. More crucially, overthinking directly leads to increased end-to-end latency, which is often the dominant factor for user satisfaction in interactive applications (Zhang et al., 2022; Roller et al., 2021). These considerations highlight the necessity of inference acceleration in a query-adaptive manner.

Several approaches have been proposed to improve reasoning efficiency, among which a practical direction is *selective thinking*: switching LRMs between non-thinking and thinking modes based on the difficulty of queries (Cheng et al., 2025; Fang et al., 2025; Li et al., 2025; Liang et al., 2025). While these methods reduce computational demands, their decision rules are often heuristic, which may route complex queries to the non-thinking model, resulting in significant performance degradation (Dekoninck et al., 2024). Such failure is particularly concerning in challenging and high-stakes settings, such as mathematical reasoning, where a small fraction of misrouted cases can dominate overall quality. Therefore, it is essential to provide efficient reasoning methods with rigorous statistical guarantees on performance loss relative to the thinking model.

To achieve reliable, efficient reasoning, *Probably Approximately Correct reasoning* (Zeng et al., 2025) provides a solution by controlling the performance loss below a user-specified level with high probability for the efficient reasoning model. However, their method is designed for offline settings and requires a calibration dataset that is i.i.d. with the test data. This is excessively restrictive for an online setting where queries arrive sequentially, and access to a calibration dataset can be impractical. Even with a calibration set, such a method often leads to uncontrolled risk or low efficiency for non-stationary data. Another significant challenge for risk control in efficient online reasoning is *partial feedback*, where potential performance loss is observed only if the thinking model is invoked. This partial observation makes standard empirical risk estimators biased, preventing the direct application of existing offline safety guarantees. Together, these motivate a foundational challenge: *how to construct an anytime-valid selection rule to switch between the thinking model and the non-thinking model.*

---

[*]Equal contribution  [1] Department of Statistics and Data Science, Southern University of Science and Technology [2] College of Computing and Data Science, Nanyang Technological University [3] School of Statistics and Data Science, Zhejiang Gongshang University [4] School of Artificial Intelligence, The Chinese University of Hong Kong, Shenzhen [5] Shenzhen Loop Area Institute. Correspondence to: Bingyi Jing <bingyijing@cuhk.edu.cn>.

*Proceedings of the 43$^{rd}$ International Conference on Machine Learning*, Seoul, South Korea. PMLR 306, 2026. Copyright 2026 by the author(s).

To address the above challenge, we propose *Betting PAC* (B-PAC) *reasoning*, a method that ensures anytime-valid performance loss control for efficient online reasoning under partial feedback. Specifically, for query $X_t$, B-PAC computes the uncertainty score of the non-thinking model $\tilde{f}(X_t)$, and accepts $\tilde{f}(X_t)$ with high probability if the score is below a carefully designed time-varying threshold $\hat{u}_{t-1}$; otherwise, the thinking model is taken for reliability. The key to the safety of B-PAC lies in modeling threshold determination as a betting game, where evidence of safety is accumulated as virtual wealth (Shafer & Vovk, 2019; Ramdas et al., 2023; Waudby-Smith & Ramdas, 2024). To address the partial observability of performance loss, B-PAC leverages inverse propensity scoring (IPS) (Horvitz & Thompson, 1952) to construct a risk estimator, based on which the wealth process is constructed. The threshold is then determined by fixed sequence testing (Angelopoulos et al., 2025), which can also be roughly regarded as the maximal value such that the associated wealth exceeds the designated safety level. In addition, we develop an adaptive betting strategy to maximize reasoning efficiency. Finally, we extend the application of B-PAC to *non-stationary data* by leveraging the technique of mixture of martingales.

In theory, we establish the distribution-free, anytime-valid performance loss control of B-PAC reasoning for both i.i.d. and non-stationary data by leveraging the tools of (super)martingales and fixed sequence testing. We also prove that the proposed adaptive betting strategy achieves logarithmic regret against the optimal fixed strategy, ensuring that the wealth process grows at a near-optimal rate to rapidly identify the most efficient safe threshold.

We extensively evaluate B-PAC reasoning across benchmarks including MATH (Hendrycks et al., 2021), MMLU-Pro (Wang et al., 2024), BIG-Bench Hard (Srivastava et al., 2023), and Magpie (Xu et al., 2025b). Results demonstrate that B-PAC reasoning outperforms existing methods by ensuring anytime-valid performance loss control while significantly reducing inference cost. For example, on Magpie with a loss tolerance of $\epsilon = 0.08$, B-PAC routes only 18.99% queries to the thinking model and achieves 41.37% token savings, while keeping the empirical loss below $\epsilon$.

We summarize our contributions as follows:

1. We propose B-PAC reasoning, a method for efficient online reasoning under partial feedback, without any access to the offline calibration dataset. To the best of our knowledge, B-PAC reasoning is the first safe, model-agnostic, efficient reasoning method in online and non-stationary settings.

2. We establish rigorous statistical properties of B-PAC reasoning, including anytime-valid performance loss control for both i.i.d. and non-stationary data, as well

as the efficiency of the threshold-updating strategy.

3. We provide comprehensive experiments on diverse reasoning benchmarks, demonstrating that B-PAC reasoning is anytime safe and efficient.

## 2. Problem Setup

Let $\mathcal{X}$ and $\mathcal{Y}$ denote the query and response spaces, respectively. We consider a framework with two LRMs: a thinking model $f : \mathcal{X} \to \mathcal{Y}$, which offers superior performance at a higher computational cost, and a non-thinking model $\tilde{f} : \mathcal{X} \to \mathcal{Y}$, which is computationally efficient but potentially less accurate. Consider the online setting where data arrive in a stream. That is, the test sample $(X_t, Y_t) \in \mathcal{X} \times \mathcal{Y}$ arrives sequentially for $t = 1, 2, \dots$, where $(X_t)_{t \geq 1}$ are independent and identically distributed, and the response $(Y_t)_{t \geq 1}$ is unobservable. Note that $Y_t$ does not need to be identically distributed. The aim of this paper is to build a sequence of more efficient composite models, denoted by $(\hat{f}_t)_{\geq 1}$, that provide probably approximately correct anytime-valid guarantees for their performance loss while improving efficiency relative to the thinking model $f$.

Formally, when $X_t$ arrives, we compute $\hat{f}_t(X_t) \in \{f(X_t), \tilde{f}(X_t)\}$, which determines whether the non-thinking model output $\tilde{f}(X_t)$ can be taken as the final answer of $X_t$ to reduce inference costs. Given an error tolerance $\epsilon > 0$ and a confidence level $1 - \alpha \in (0, 1)$, our goal is to determine composite models $(\hat{f}_t)_{t \geq 1}$ that always control the performance loss below the level $\epsilon$ at any time $t$ with probability at least $1 - \alpha$, while improving efficiency as much as possible. We formalize the anytime $(\epsilon, \alpha)$-PAC efficiency as follows.

**Definition 2.1** (Anytime $(\epsilon, \alpha)$-PAC efficiency). A sequence of models $(\hat{f}_t)_{t \geq 0}$ is said to be anytime $(\epsilon, \alpha)$-PAC efficient with respect to a loss $l \in [0, 1]$ if, for any given $\epsilon > 0$ and $\alpha \in (0, 1)$,

$$\mathbb{P}(\forall t \geq 1 : R_t(\hat{f}_t) \leq \epsilon) \geq 1 - \alpha, \tag{1}$$

where $R_t(\hat{f}_t) = \mathbb{E}_{X \sim P_X}[l(\hat{f}_t(X), f(X))]$ is the risk function at time $t$, measuring the performance loss with respect to the thinking model, $l(\cdot, \cdot)$ is a loss function, and $X \sim P_X$. The probability is taken over the randomness of the streaming data and internal randomization of the procedure that generates $(\hat{f}_t)_{t \geq 1}$.

*Remark* 2.2 (Data Assumption). The i.i.d. assumption of $X_t$ is mainly adopted to present the B-PAC reasoning method more clearly and to leverage fixed sequence testing to obtain a highly efficient model $\hat{\tilde{f}}_t$. In Section 5, we show that the B-PAC reasoning method can handle *non-stationary* data.

*Remark* 2.3 (Loss Function). B-PAC reasoning works for any bounded loss function, such as 0-1 loss for verifiable

tasks. We assume $l(\cdot, \cdot) \in [0, 1]$ without loss of generality, since any bounded loss can be scaled to this interval. Note that performance loss is measured relative to $f$, not to the ground truth $Y_t$, which is reasonable and necessary in efficient reasoning; see Appendix A for a detailed discussion.

## 3. Betting PAC Reasoning

This section introduces the method of B-PAC reasoning in three steps. We first introduce its working principle under a given threshold, and then explain how to update the threshold through the betting process. Finally, we provide an adaptive betting strategy to maximize reasoning efficiency.

### 3.1. Uncertainty-based Routing Mechanism

At time $t$, we first use the non-thinking model $\tilde{f}$ to produce $\tilde{f}(X_t)$, and compute its uncertainty score $U_t \in [0, 1]$. For example, we can use verbalized confidence scores (Xiong et al., 2024; Tian et al., 2023; Yang et al., 2024) from a non-thinking model or scores based on prediction logits (Hao et al., 2023; Huang et al., 2025). In line with intuition, these uncertainty scores should be positively correlated with the likelihood of inconsistency with the thinking model $f$. Therefore, if $U_t$ is small, B-PAC reasoning tends to take $\tilde{f}(X_t)$ as a proxy of $f(X_t)$. For $\tilde{f}(X_t)$ with large $U_t$, B-PAC reasoning invokes the thinking LRM to produce $f(X_t)$.

Formally, denote the calculated threshold at time $t - 1$ by $\hat{u}_{t-1}$ (the construction will be introduced later). Given the test point $X_t$ and threshold $\hat{u}_{t-1}$, B-PAC calls thinking model with probability of $\pi_t$, where

$$\pi_t = \pi(U_t; \hat{u}_{t-1}, \rho_t) = \mathbb{I}\{U_t \geq \hat{u}_{t-1}\} + \rho_t \mathbb{I}\{U_t < \hat{u}_{t-1}\},$$

where $\rho_t \in (0, 1)$ is a minimum exploration probability at time $t$, which can be either a constant or a time-varying parameter determined by a specific strategy; see Section 3.3. More specifically, let $\xi_t \sim \text{Bernoulli}(\pi_t)$. The composite model is constructed by

$$\hat{f}_t(X_t) = (1 - \xi_t)\tilde{f}(X_t) + \xi_t f(X_t). \tag{2}$$

For $U_t \geq \hat{u}_{t-1}$, we have $\hat{f}_t(X_t) = f(X_t)$. If $U_t < \hat{u}_{t-1}$, then $\hat{f}_t(X_t) = \tilde{f}(X_t)$ holds with a large probability $1 - \rho_t$.
*Remark* 3.1 (Uncertainty Score). Practitioners can utilize various existing uncertainty scores since the safety of B-PAC holds for any type of score; see Section 4. But to gain reasoning acceleration, scores $U_t$ correlated with the likelihood of disagreement with $f(X_t)$ are preferred. For detailed discussions, see Appendix A.

### 3.2. Threshold Selection as a Betting Process

It remains to provide the strategy of constructing adaptive threshold $\{\hat{u}_t\}_{t=1}^{\infty}$, which is a key step to achieve the anytime $(\epsilon, \alpha)$-PAC efficiency. At a high level, we achieve

this goal by (a) constructing an IPS estimator for the realized risk; (b) constructing a supermartingale based on IPS estimator; and (c) leveraging the fixed sequence testing.

**Step 1: IPS estimator.** Denote the loss of the non-thinking model by $l_t = l(\tilde{f}(X_t), f(X_t))$, which is observable only if the thinking model is utilized (i.e., $\xi_t = 1$). Note that $l_t$ is not the realized loss $l(\hat{f}_t(X_t), f(X_t))$ of B-PAC reasoning. To estimate the risk associated with threshold $u$ given partial feedback, we construct the IPS estimator for the risk by

$$Z_t(u) = (1 - \rho_{\min})\frac{l_t}{\pi_t}\xi_t \mathbb{I}\{U_t < u\}, \tag{3}$$

where $\rho_{\min} = \inf_{t \geq 1} \rho_t$. Here, the term $\xi_t/\pi_t$ corrects the selection bias inherent in the partial feedback while the coefficient $(1 - \rho_{\min})$ accounts for the time-varying $\rho_t$, jointly ensuring that $Z_t(u)$ serves as a tight upper bound of the true risk under threshold $u$.

**Step 2: Supermartingale.** Denote the filtration $\mathcal{F}_t$ by the $\sigma$-algebra generated by observations up to time $t$, i.e.,

$$\mathcal{F}_t = \sigma(\{(X_i, \tilde{f}(X_i), l_i\xi_i, U_i)\}_{i=1}^t). \tag{4}$$

Intuitively, $\mathcal{F}_t$ represents all information available before time $t + 1$. For $\epsilon \in (0, 1)$ and $u \in [0, 1]$, let

$$D_t(u) = \epsilon - Z_t(u). \tag{5}$$

Set $K_0(u) = 1$. We construct a process $(K_t(u))_{t \geq 0}$ adapted to $(\mathcal{F}_t)_{t \geq 0}$ by

$$K_t(u) = K_{t-1}(u)(1 + \lambda_t(u)D_t(u)), \tag{6}$$

where $\lambda_t(u) \in \mathcal{F}_{t-1}$ is a non-negative random variable satisfying $1 + \lambda_t(u)D_t(u) \geq 0$. In Section 4, we show that $(K_t(u))_{t \geq 0}$ is a nonnegative supermartingale (Lemma 4.1).

**Step 3: Threshold selection.** We complete the description of B-PAC reasoning by introducing how to update threshold $\hat{u}_t$ based on $K_t$. Denote the search space of threshold by $\mathcal{U} = \{u^{(1)}, \ldots, u^{(N)}\}$, where $0 = u^{(1)} < u^{(2)} < \cdots < u^{(N)} = 1$. B-PAC reasoning determines the threshold $\hat{u}_t$ by

$$\hat{u}_t = \max\left\{u^{(i)} \in \mathcal{U} : \forall j \leq i, K_t(u^{(j)}) \geq \frac{1}{\alpha}\right\}, \tag{7}$$

where the maximum is to obtain a most efficient model, since a large value $\hat{u}_t$ leads to a higher probability that $\xi_{t+1} = 0$. If $\{u^{(i)} \in \mathcal{U} : \forall j \leq i, K_t(u^{(j)}) \geq 1/\alpha\} = \emptyset$, we set $\hat{u}_t = 0$. The motivation behind (7) relies on our insights to Ville's inequality (Howard et al., 2020) and fixed sequence testing (Angelopoulos et al., 2025); see the proof of Theorem 4.2 in Appendix B. Here, we provide an intuitive explanation for $K_t(u)$ and $\hat{u}_t$ as follows.

**Betting explanation.** We interpret $(K_t(u))_{t \geq 0}$ as the accumulated capital of a gambler testing the null hypothesis that threshold $u$ is unsafe. In this game, the predictor $\lambda_t \in \mathcal{F}_{t-1}$ acts as the wager, determined prior to round $t$. The term $D_t(u)$ represents the payoff of each unit, comprising the risk tolerance $\epsilon$ against the estimated loss $Z_t(u)$. Under the null hypothesis (unsafe threshold $u$), $(K_t(u))_{t \geq 0}$ forms a supermartingale, implying that the expected wealth decreases over time, i.e., $\mathbb{E}[K_t(u)|\mathcal{F}_{t-1}] \leq K_{t-1}(u)$. Conversely, if $u$ is truly safe, the capital is expected to grow. Consequently, a large value of $K_t(u)$ serves as strong evidence of safety, motivating the selection rule in (7).

*Remark* 3.2 (Non-trivial setting). We only need to consider the case that $\epsilon < 1 - \rho_t$ such that $D_t(u) < 0$ when $\xi_t \mathbb{I}\{U_t < u\} = 1$. For $\epsilon \geq 1 - \rho_t$, the risk control becomes trivial.

### 3.3. Hyperparameter Selection

The choices of $(\rho_t)_{t \geq 0}$ and $(\lambda_t)_{t \geq 1}$ do not influence the anytime $(\epsilon, \alpha)$-PAC efficient guarantee (see Theorem 4.2), but can have an impact on the reasoning efficiency. We first provide an adaptive betting strategy of $(\lambda_t)_{t \geq 1}$ as follows.

**Choice of $\lambda_t$.** To gain maximum efficiency, the predictor $\lambda_t(u)$ should maximize wealth $K_t(u)$, or equivalently, maximize $\log K_t(u)$. By (5), we have $D_t(u) \in [\epsilon - (1 - \rho_{\min})/\rho_t, \epsilon]$. Under the constraint that $\lambda_t \geq 0$ and $1 + \lambda_t(u)D_t(u) \geq 0$, we have $\lambda_t \in [0, 1/((1-\rho_{\min})/\rho_t - \epsilon)]$. To avoid the worst-case scenario $K_t = 0$, we consider $\lambda_t \in [0, c/((1 - \rho_{\min})/\rho_t - \epsilon)]$, where $c \in (0, 1)$ (usually $c$ is close to 1). Note that

$$\frac{d}{d\lambda} \log K_t(u) \approx \sum_{i=1}^{t-1} (D_i(u) - D_i^2(u)\lambda_i(t)).$$

By optimization theorem (Hazan, 2016), we choose

$$\lambda_t(u) = \min\left\{\max\left\{\frac{\sum_{i=0}^{t-1} D_i(u)}{\sum_{i=0}^{t-1} D_i^2(u) + 1}, 0\right\}, \frac{c}{M_t}\right\}, \tag{8}$$

where $M_t = \max\{\epsilon, (1 - \rho_{\min})/\rho_t - \epsilon\}$.

**Choice of $\rho_t$.** The value of $\rho_t$ impacts efficiency in two ways. First, a small $\rho_t$ leads to a large value of $Z_t(u)$ if $\xi_t \mathbb{I}\{U_t < u\} = 1$, which forces a conservative betting strategy ($\lambda_t \leq c/(1/\rho_t - \epsilon) \approx 0$) to maintain non-negative wealth. This limits the wealth growth rate even when the threshold is safe. In contrast, a large $\rho_t$ permits an aggressive betting strategy, which accelerates the identification of optimal thresholds during the initial phase. Second, when $\hat{u}_t$ stabilizes, a large $\rho_t$ will lead to a low efficiency since we call the thinking model with at least $\rho_t$ probability.

To strike a balance between fast convergence (warm-up) and long-term efficiency (deployment), we recommend a

---

**Algorithm 1** Betting PAC Reasoning

1: **Input:** Queries $(X_t)_{t \geq 1}$; thinking model $f$, non-thinking model $\tilde{f}$, loss function $l$, error tolerance $\epsilon$, confidence level $\alpha$, minimum exploration probabilities $(\rho_t)_{t \geq 1}$ given by (9), search space $\mathcal{U}$.
2: **Output:** response sequence $(\hat{f}_t(X_t))_{t \geq 1}$.
   **Initialize:** $\hat{u}_0 = 0$, $K_0 = 1$.
3: **for** $t = 1, 2, \ldots$ **do**
4:    Compute $\tilde{f}(X_t)$ and uncertainty score $U_t$.
5:    Compute $\pi_t = \mathbb{I}\{U_t \geq \hat{u}_{t-1}\} + \rho_t \mathbb{I}\{U_t < \hat{u}_{t-1}\}$.
6:    Sample $\xi_t \sim \text{Bernoulli}(\pi_t)$.
7:    **if** $\xi_t = 0$ **then**
8:       $\hat{f}(X_t) \leftarrow \tilde{f}(X_t)$
9:    **else**
10:      $\hat{f}(X_t) \leftarrow f(X_t)$
11:   **end if**
12:   Compute $Z_t(u)$ as in (3), $u \in \mathcal{U}$.
13:   Compute $D_t(u) = \epsilon - Z_t(u)$, $u \in \mathcal{U}$.
14:   Compute the predictor $\lambda_t(u)$ as in (8), $u \in \mathcal{U}$.
15:   Update $K_t(u) = K_{t-1}(u)(1 + \lambda_t(u)D_t(u))$, $u \in \mathcal{U}$.
16:   Update the threshold $\hat{u}_t$ as in (7).
17: **end for**
18: **Return** $(\hat{f}_t(X_t))_{t \geq 1}$.

---

two-stage exploration strategy that

$$\rho_t = \rho_{\text{warm}}\mathbb{I}\{t \leq T_{\text{warm}}\} + \rho_{\text{deploy}}\mathbb{I}\{t > T_{\text{warm}}\}, \tag{9}$$

where $\rho_{\text{warm}}$ and $\rho_{\text{deploy}}$ take suitable large and small values, respectively, and $T_{\text{warm}}$ is a hyperparameter. For this strategy, we have $\rho_{\min} = \rho_{\text{deploy}}$. The B-PAC reasoning method is summarized in Algorithm 1.

## 4. Theoretical Analysis

In this section, we establish the theoretical foundations of B-PAC reasoning. We first establish that $(\hat{f}_t)_{t \geq 1}$ given by Algorithm 1 is anytime $(\epsilon, \alpha)$-PAC efficient. Then we provide the efficiency of the betting strategy given by (8). All technical proofs are presented in Appendix B. For brevity, by (2), we re-parameterize the risk $R_t(\hat{f}_t)$ as $R_t(\hat{u}_{t-1})$ with a slight abuse of notation. Specifically, for fixed $u \in \mathcal{U}$, we have $R_t(u) = \mathbb{E}[l_t \mathbb{I}\{\xi_t(u) = 0\}]$, where $\xi_t(u) \sim \text{Bernoulli}(\mathbb{I}\{U_t \geq u\} + \rho_t \mathbb{I}\{U_t < u\})$.

### 4.1. Safety of B-PAC Reasoning

Consider the i.i.d. setting. By Lemma B.1, we have $R_t(u) = (1 - \rho_t)\mathbb{E}[l_t \mathbb{I}\{U_t < u\}]$. Let $(\rho_t)_{t \geq 1}$ given by (9) and denote the *deployment risk* with threshold $u$ by

$$R(u) = (1 - \rho_{\text{deploy}})\mathbb{E}[l_t \mathbb{I}\{U_t < u\}].$$

Define the null hypothesis by

$$H_{0,u} : R(u) > \epsilon.$$

We begin by showing that for $u \in \mathcal{U}$, $(K_t(u))_{t \geq 0}$ (6) is a nonnegative supermartingale under $H_{0,u}$.

**Lemma 4.1** (Supermartingale Property). *Let $\mathcal{F}_t$, $D_t(u)$, and $\rho_t$ be defined by (4), (5), and (9), respectively. Under the null hypothesis $H_{0,u}$, for any nonnegative $\lambda_t(u) \in \mathcal{F}_{t-1}$ satisfying $1 + \lambda_t(u)D_t(u) \geq 0$, the process $(K_t(u))_{t \geq 0}$ (6) is a non-negative supermartingale with respect to $\mathcal{F}_t$.*

We observe that $R(u)$ exhibits inherent monotonicity: as $u$ increases, the condition $\mathbb{I}\{U_t < u\}$ is satisfied more frequently, leading to a non-decreasing risk. This monotonicity allows us to leverage Lemma 4.1 and fixed sequence testing to derive the safety guarantee.

**Theorem 4.2** (Anytime $(\epsilon, \alpha)$-PAC efficient). *Let $(K_t(u))_{t \geq 0}$, $\hat{u}_t$, and $(\rho_t)_{t \geq 0}$ be defined by (5), (7), and (9), respectively. For any $\alpha \in (0,1)$, $\epsilon \in (0,1)$, and any nonnegative $\lambda_t \in \mathcal{F}_{t-1}$ with $1 + \lambda_t D_t(u) \geq 0$, B-PAC reasoning satisfies that*

$$\mathbb{P}(\forall t \in \mathbb{N} : R_t(\hat{u}_t) \leq \epsilon) \geq 1 - \alpha. \quad (10)$$

### 4.2. Efficiency of B-PAC Reasoning

Although the safety of the B-PAC reasoning holds regardless of the choice of $\lambda_t$, how quickly it identifies the tightest threshold depends on the growth rate of wealth $K_t(u)$. For any fixed $u \in \mathcal{U}$, this objective can be restated as maximizing $\log K_T(u) = \sum_{t=1}^{T} \log(1 + \lambda D_t(u))$ for $T \in \mathbb{N}$. To achieve a computationally efficient strategy with closed-form updates, we follow the standard approach of maximizing a quadratic surrogate. Based on the second-order Taylor expansion $\log(1+x) \approx x - x^2/2$, we define the quadratic proxy for the wealth growth at time $t$ by

$$g_t(\lambda; u) = \lambda D_t(u) - \frac{1}{2}\lambda^2 D_t^2(u).$$

Since $\lambda_t \in [0, c/((1-\rho_{\min})/\rho_t - \epsilon)]$, we denote $\lambda_T^*$ by

$$\lambda_T^*(u) = \arg \max_{0 \leq \lambda \leq c/((1-\rho_{\min})/\rho_T - \epsilon)} \sum_{t=1}^{T} g_t(\lambda; u). \quad (11)$$

The following theorem establishes the efficiency of the proposed strategy $(\lambda_t(u))_{t \geq 0}$.

**Theorem 4.3.** *Let $\lambda_t(u)$, $\rho_t$, and $\lambda_T^*$ be defined by (8), (9), and (11), respectively. Define the regret of strategy $(\lambda_t(u))_{t \geq 0}$ by $\mathcal{R}_T^{quad} = \sum_{t=1}^{T}(g_t(\lambda_T^*(u)) - g_t(\lambda_t(u)))$. For $T > T_{warm}$, we have*

$$\mathcal{R}_T^{quad} \leq \frac{c^2}{2M^2} + \frac{(1+c)^2 M^2}{2\beta \log(1+M^2)} \log(TM^2 + 1),$$

*where $M = \max\{\epsilon, 1/\rho_{deploy} - (1+\epsilon)\}$.*

Therefore, we have $\mathcal{R}_T^{quad} = O(\log T)$, which means that B-PAC can match the optimal rate. Consequently, B-PAC can rapidly converge to the optimal efficient threshold, thereby maximizing the reasoning efficiency.

## 5. Extensions to Non-Stationary Data

In this section, we extend the method of B-PAC reasoning to handle potential non-stationary data $(X_t)_{t \geq 0}$, such as data with distribution shifts, temporal drifts, or even adversarial inputs. These settings are common in real-world deployment such as multi-turn dialogue scenarios.

### 5.1. B-PAC Reasoning for Non-Stationary Data

By modifying the equation (7) of Step 3, the framework of B-PAC reasoning introduced in Section 3 also holds for non-stationary data. Specifically, let $\nu(u)$ be the prior probability mass for threshold $u$ with $\sum_{u \in \mathcal{U}} \nu(u) = 1$. The data-dependent threshold $\hat{u}_t$ at time $t$ is determined by

$$\hat{u}_t = \max \left\{ u \in \mathcal{U} : K_t(u) \geq \frac{1}{\alpha\nu(u)} \right\}, \quad (12)$$

where $K_t(u)$ is defined the same as (6). If $\{u \in \mathcal{U} : K_t(u) \geq 1/\alpha\nu(u)\} = \emptyset$, we set $\hat{u}_t = 0$.

The prior $\nu(u)$ enables the incorporation of prior knowledge to improve efficiency. For example, a large probability mass can be allocated to a relatively large threshold $u \in \mathcal{U}$ if we have confidence in the accuracy of the non-thinking model $\tilde{f}$ and the quality of uncertainty score. This can result in a large $\hat{u}_t$, which then improves efficiency. However, in most cases, B-PAC reasoning with (12) is less efficient than the one with fixed sequence testing (7) since $0 < \nu(u) \leq 1$, which also reflects a fundamental safety-efficiency trade-off.

### 5.2. Safety for Non-Stationary Data

Let $(\mathcal{F}_t)_{t \geq 0}$ be the filtration defined by (4). For $u \in \mathcal{U}$, define the weighted cumulative risk at time $t$ by

$$L_t(u) = \sum_{j=1}^{t} \frac{\lambda_j(u)}{\sum_{i=1}^{t} \lambda_i(u)} \mathbb{E}[l_j \mathbb{I}\{\xi_j(u) = 0\}|\mathcal{F}_{j-1}], \quad (13)$$

where $\xi_j(u) \sim \text{Bernoulli}(\mathbb{I}\{U_j \geq u\} + \rho_t \mathbb{I}\{U_j < u\})$. For fixed betting strategies, it degenerates into non-weighted risk. For the i.i.d. case, we have $L_t(u) = R_t(u)$. The following theorem demonstrates the safety of B-PAC reasoning for non-stationary data.

**Theorem 5.1** (Safety). *Consider that $(X_t)_{t \geq 0}$ is an arbitrary data stream. Let $(K_t(u))_{t \geq 0}$ and $\hat{u}_t$ defined by (5) and (12), respectively. For any prior probability mass $\nu$ with domain $\mathcal{U}$, any $\alpha \in (0,1)$, $\epsilon \in (0,1)$, and any $\rho_t \in (0,1)$, $\lambda_t \in \mathcal{F}_{t-1}$ with $1 + \lambda_t D_t(u) \geq 0$, we have*

$$\mathbb{P}(\forall t \in \mathbb{N} : L_t(\hat{u}_t) \leq \epsilon) \geq 1 - \alpha.$$

Theorem 5.1 can be regarded as a safety guarantee for the worst cases, since it holds for any types of non-stationary data. This implies that a tighter bound may be derived under some mild assumptions about the data $(X_t)_{t\geq1}$. In practical deployment, we suggest simply employing the update rule given by (7) to gain high efficiency, since empirically it also leads to valid risk control for non-stationary data; see Figures 1 and 2.

# 6. Experiments

In this section, we present the experimental results of B-PAC reasoning and other baselines on diverse benchmarks. We aim to demonstrate that B-PAC reasoning always controls risk with a specific confidence level for streaming data while significantly reducing computational overhead. The reproduction codes can be found at `https://github.com/ChengyaoYu1/B-PAC-Reasoning`.

## 6.1. Experimental Setup

We introduce the main information about the experiments; detailed settings and additional results such as results on other uncertainty scores and ablation studies on the proposed strategies of $\lambda_t$ and $\rho_t$ are presented at Appendix C and D.

**Datasets.** We evaluate B-PAC reasoning across a diverse suite of benchmarks that encompass mathematical problem-solving, multi-task knowledge, symbolic reasoning, and open-ended instruction following. Specifically, we leverage four challenging datasets: MATH (Hendrycks et al., 2021), MMLU-Pro (Wang et al., 2024), BIG-Bench Hard (BBH) (Srivastava et al., 2023), and Magpie (Xu et al., 2025b). To ensure computational feasibility given the scale of our evaluation, we use randomly sampled subsets of MMLU-Pro and Magpie in our experiments.

**Large language models.** We utilize the Qwen3 model family (Yang et al., 2025). Specifically, we employ `Qwen3-4B-Thinking-2507` as the *thinking model* and `Qwen3-4B-Instruct-2507` as the *non-thinking model*. We configure the sampling temperature and other hyperparameters as the settings in the original technical report.

**Methods.** We first compare B-PAC with the offline method *PAC reasoning* (Zeng et al., 2025). For both methods, we use logits-based uncertainty score. We then compare B-PAC with two online efficient methods: *O-naive* and *IPS+Hoeff*, where O-naive has no safety guarantee and IPS+Hoeff provides a safety guarantee by leveraging Hoeffding's inequality (Hoeffding, 1963). Other efficient methods including *Chain of Draft* (CoD) (Xu et al., 2025a) and *NoThinking* (Ma et al., 2025) are also compared.

**Loss functions.** For verifiable tasks, we use 0-1 loss. For open-ended tasks, we use a loss function based on an LLM-as-a-judge. For dataset including multi-type tasks, we adopt a piecewise loss function.

**Evaluation.** We evaluate reasoning efficiency through two metrics: *Expert Call Percentage* (ECP) and *Token Percentage* (TP). For $t \geq 1$, we define

$$\text{ECP}_t = \frac{1}{t} \sum_{i=1}^{t} \mathbb{I}\{\xi_i = 1\} \times 100\%,$$

$$\text{TP}_t = \frac{\sum_{i=1}^{t}(\tilde{h}(X_i) + h(X_i)\mathbb{I}\{\xi_i = 1\})}{\sum_{i=1}^{t} h(X_i)} \times 100\%.$$

where $\tilde{h}(x_i)$ and $h(x_i)$ represent the number of tokens of $\tilde{f}(x_i)$ and $f(x_i)$, respectively. The safety is evaluated by the *Empirical Risk* (ER):

$$\text{ER}_t = \frac{1}{t} \sum_{i=1}^{t} l(\hat{f}_i(X_i), f(X_i)).$$

*Remark* 6.1 (ECP vs TP). With $h(X_i)/\tilde{h}(X_i) = S$, we have $\text{TP}_t = \text{ECP}_t + 1/S$. Since $S$ depends on $(f, \tilde{f})$, we regard ECP as a more intrinsic metric for evaluating the efficiency of routing methods, as it isolates the decision-making quality from the specific model characteristics.

## 6.2. Results

**Efficiency outperforms offline method.** Figure 1 compares the performance of B-PAC against the offline PAC baseline. We observe that B-PAC consistently maintains the empirical risk below the tolerance $\epsilon$ while significantly reducing computational costs, achieving ECP $= 18.99\%$ and TP $= 58.63\%$. In contrast, PAC exhibits negligible efficiency gains. This disparity arises because PAC relies on a fixed routing threshold derived from a calibration set from a harder dataset, BBH. Consequently, when the test stream introduces easier queries, the fixed threshold remains overly conservative. B-PAC, however, dynamically relaxes the threshold in response to the easier streaming data, thereby maximizing efficiency without compromising safety.

**Risk control under non-stationary settings.** Figure 2 illustrates performance under a dynamic setting where query difficulty escalates over time. The offline PAC baseline fails to control risk, as its fixed threshold—calibrated on earlier, easier data—cannot adapt to the harder incoming queries. Conversely, B-PAC demonstrates superior robustness, achieving anytime-valid safety. This is attributed to B-PAC's ability to dynamically tighten the routing threshold as the error rate rises, effectively trading off necessary computation for strict adherence to the safety constraint.

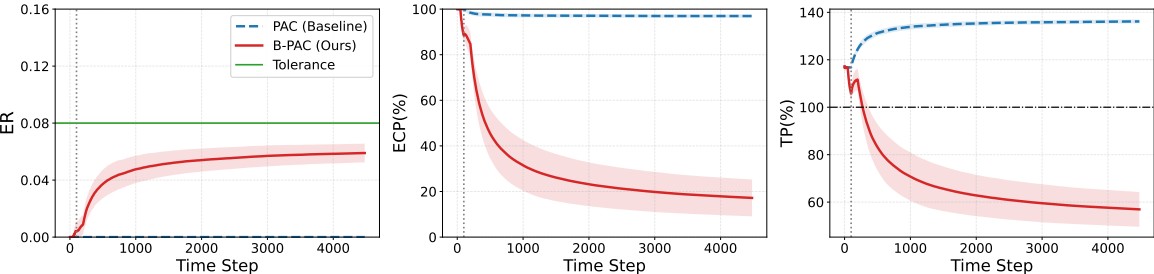

*Figure 1.* **Efficiency outperforms offline PAC reasoning**. ER, ECP, and TP are reported on a combined dataset of Magpie and BBH, with $\epsilon = 0.08$ and $\alpha = 0.1$. The vertical dotted line indicates the size of the calibration set used for the offline PAC baseline. Experiments are repeated 100 times, and the shaded areas represent standard deviations.

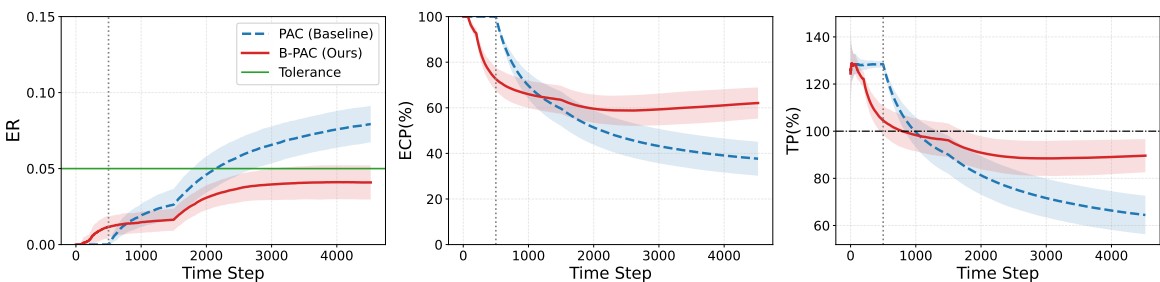

*Figure 2.* **Anytime safety for non-stationary data**. Results are reported on a combined dataset of MMLU-Pro and BBH, with $\epsilon = 0.05$ and $\alpha = 0.1$.

*Table 1.* Results of B-PAC reasoning, CoD, and NoThinking on MATH and MMLU-Pro, with $\epsilon = 0.08$ and $\alpha = 0.1$.

| Metric | MATH | | | MMLU-Pro | | |
|---|---|---|---|---|---|---|
| | **B-PAC** | **CoD** | **NoThinking** | **B-PAC** | **CoD** | **NoThinking** |
| ER ↓ | **0.03 ± 0.01** | 0.1204 | 0.1577 | **0.03 ± 0.01** | 0.10 | 0.12 |
| TP (%) ↓ | **68.16 ± 14.56** | 95.63 | 77.26 | 77.24 ± 9.94 | **73.63** | 76.52 |

**Comparison with online methods.** Figure 3 benchmarks B-PAC against O-naive and IPS+Hoeff on MMLU-Pro and BBH, which highlights the critical challenges in online efficient reasoning. The failure of O-naive (risk violation) confirms that handling partial feedback is non-trivial and requires accurate estimation like IPS. Meanwhile, the inefficiency of IPS+Hoeff (near 100% expert usage) demonstrates that standard inequalities are too loose for practical deployment. B-PAC overcomes both limitations by leveraging IPS estimation with a tight betting supermartingale, achieving guaranteed safety without sacrificing efficiency. For example, for MMLU-Pro, B-PAC reasoning achieves ECP = 47.04% and TP = 78.07%.

**Reliability against heuristic approaches.** We extend our comparison to heuristic reasoning paradigms, specifically CoD and NoThinking; see Table 1. The results reveal that heuristic baselines violate the risk tolerance ER > 0.08, failing to provide safety guarantees. In contrast, B-PAC

uniquely maintains risk control due to its rigorous theoretical foundation. Crucially, this safety does not come at the cost of efficiency; B-PAC achieves the lowest token consumption on the complex MATH dataset (TP = 68.16%) and remains competitive on MMLU-PRO, establishing it as the only reliable solution for high-stakes deployment.

## 7. Related Work

**Anytime safe efficient reasoning.** Although LRMs have demonstrated remarkable progress in tackling complex tasks (Sprague et al., 2024; Bai et al., 2025), the problem of overthinking makes the usage of LRMs expensive and even dangerous (Chen et al., 2024; Sui et al., 2025; Cuadron et al., 2025). This highlights the need to improve the inference efficiency of LRMs. By Sui et al. (2025), recent efficient reasoning methods include three categories: model-based efficient reasoning, reasoning output-based efficient reasoning, and input prompts-based efficient reasoning. Among them, a promising direction is to switch between thinking and non-thinking models to save computational overhead (Cheng et al., 2025; Fang et al., 2025; Ma et al., 2025; Li et al., 2025; Liang et al., 2025; Paliotta et al., 2025; Chung et al., 2025; Pan et al., 2024; Yong et al., 2025). However, these methods cannot control the performance loss caused by using a non-thinking model. The proposed B-PAC reason-

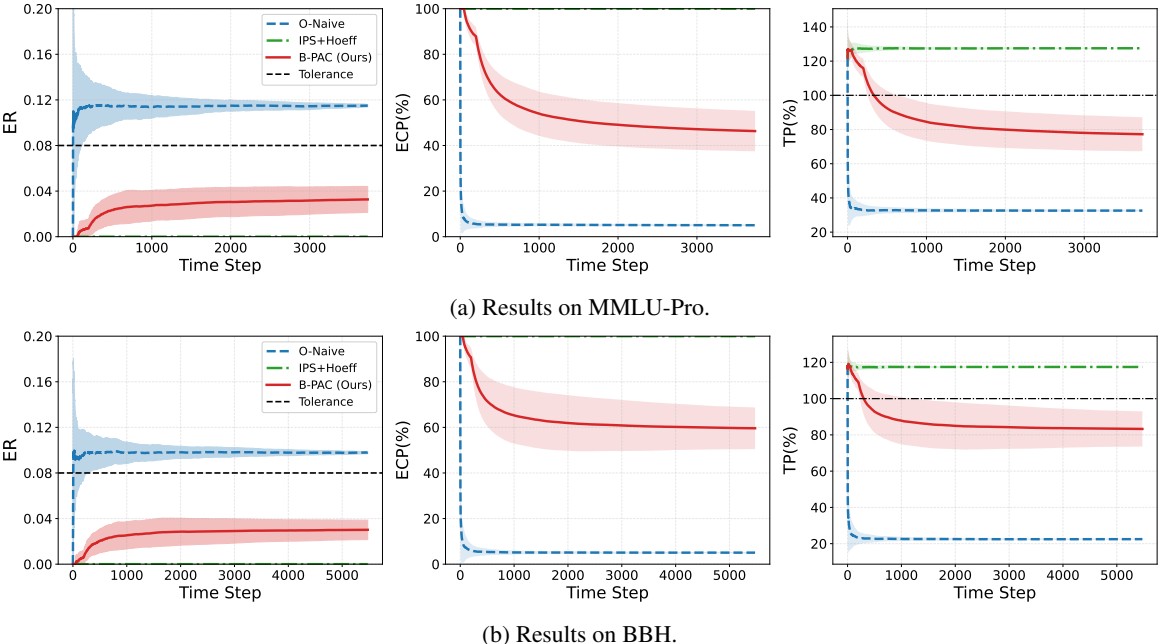

(a) Results on MMLU-Pro.

(b) Results on BBH.

*Figure 3.* **Safety and efficiency of B-PAC reasoning compared with online methods, including IPS+Hoeff and O-Naive.** Results are reported on MMLU-Pro and BBH benchmarks, with $\epsilon = 0.08$ and $\alpha = 0.1$.

ing fills this gap, providing a model-agnostic, anytime safe method for improving reasoning efficiency.

Game-theoretic statistics, or more specifically, the non-negative (super)martingale and e-process provide fundamental tools for safe anytime-valid inference (Shafer, 2021; Ramdas et al., 2020; 2023; Chugg et al., 2023; Waudby-Smith & Ramdas, 2024), where "e" of e-process refers to e-values (Shafer, 2021; Vovk & Wang, 2021; Yu et al., 2024). The main idea of these tools is to construct a powerful test (super)martingale (or e-process) and then leverage Ville's inequality (Howard et al., 2020). Our work introduces these tools to efficient reasoning and takes additional techniques to address specific challenges in this field. First, we tackle the partial feedback by integrating the IPS estimator into the supermartingale construction. Second, for i.i.d. settings, we innovatively combine fixed sequence testing with Ville's inequality, which is more efficient than directly leveraging the tools of mixture martingales. Furthermore, we formulate the betting strategy as an online convex optimization problem. These contributions enable B-PAC reasoning to be an anytime safe and efficient reasoning method.

**Risk control and PAC reasoning.** Providing model-agnostic risk control for black-box models has become a prominent research direction in recent years, such as methods of conformal prediction (Vovk et al., 2005; Bates et al., 2021; Angelopoulos et al., 2024; Angelopoulos & Bates, 2023), Learn then Test (LTT) (Angelopoulos et al., 2025), and a predictive inference style for PAC learning (Valiant, 1984; Candès et al., 2025; Zeng et al., 2025). The key insight

of these methods lies in leveraging a hold-out calibration set to estimate the empirical quantile (or other statistics) of the risk distribution, thereby determining a fixed threshold that guarantees the level of errors on the test data remains within a user-specified level. Among these methods, the most related work is the PAC reasoning (Zeng et al., 2025), which first applies the LTT to improve the efficiency of reasoning models. Specifically, PAC reasoning determines a fixed switching threshold on a calibration set by constructing an upper confidence bound (UCB) on the performance loss and leveraging LTT (Angelopoulos et al., 2025; Candès et al., 2025). Our proposed B-PAC reasoning fundamentally differs from (Zeng et al., 2025) in two aspects:

1. Zeng et al. (2025) target offline settings with i.i.d calibration data. In contrast, B-PAC is designed for online settings with partial feedback, does not require access to offline data, and is applicable to non-stationary data.

2. Zeng et al. (2025) depend on the tools of UCB and LTT. In contrast, to tackle partial feedback and online settings, B-PAC reasoning is mainly built on the construction of the IPS estimator and supermartingale. This also leads to a distinct safety guarantee.

# 8. Conclusion

We propose B-PAC reasoning, a rigorous method designed to bridge the gap between reasoning efficiency and safety for the online deployment of LRMs. Specifically, B-PAC reasoning switches a query to the thinking mode only when

its uncertainty level exceeds the current threshold, where the determination of the time-varying threshold is formulated as a betting game. Unlike existing heuristic-motivated and offline selective thinking methods, B-PAC ensures anytime-valid performance loss control even under the challenging settings of partial feedback and non-stationary data. We further establish the efficiency of our proposed adaptive betting strategy, by proving it achieves logarithmic regret. Extensive experiments demonstrate that B-PAC reasoning can significantly reduce computational costs and control the risk below the user-specified level. We expect that the proposed method can serve as a basic tool in the area of safe efficient reasoning.

**Broader applications.** The proposed B-PAC is model-agnostic: no assumptions are imposed for $f$, $\tilde{f}$, and the uncertainty score $U$. Therefore, it is promising to apply B-PAC to other routing systems with a cost-accuracy trade-off.

**Limitation.** Although B-PAC reasoning provides the first theoretically grounded method for safe efficient reasoning, several challenges remain to be solved to expand its usability. First, the present B-PAC reasoning switches between two models, extending it to multiple models remains unsolved. Second, the efficiency of B-PAC relies on the quality of uncertainty scores. When multiple scores are available, it is unknown how to adaptively select the most reliable score. Third, designing the conditional case (Zeng, 2025) of B-PAC reasoning may further enhance the efficiency.

## Acknowledgements

We thank the anonymous reviewers for their valuable comments and suggestions. This research is supported by the SUSTech-NUS Joint Research Program.

## Impact Statement

This paper presents work whose goal is to advance the field of Machine Learning. There are many potential societal consequences of our work, none which we feel must be specifically highlighted here.

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

# A. Deferred Discussions

## A.1. Rationality for Controlling Performance Loss with respect to Thinking Model

Recall that the primary goal of B-PAC reasoning is to control the performance loss with respect to the thinking model $f(X_t)$, rather than the ground truth $Y_t$. While controlling risk relative to the ground truth is an ultimate goal for reliable AI, we justify the rationality and necessity of the goal of B-PAC reasoning from the following three perspectives.

**Relationship with controlling performance loss with respect to ground truth.** The performance loss of an efficient reasoning model $\hat{f}_t$ relative to the ground truth $Y_t$ can be decomposed into two components:

$$\underbrace{l(\hat{f}_t(X_t), Y_t)}_{\text{Total Error}} \leq \underbrace{l(\hat{f}_t(X), f(X_t))}_{\text{Approximation Error}} + \underbrace{l(f(X_t), Y_t)}_{\text{Inherent Error}}.$$

The *approximation error* stems from using the non-thinking model $\tilde{f}$ instead of the thinking model $f$, which is introduced by the efficiency mechanism and is exactly what B-PAC reasoning aims to control. The *inherent error* stems from the limitations of the thinking model $f$ itself. Reducing the inherent error falls within the scope of model alignment or pretraining, which is orthogonal to the problem of efficient reasoning. Furthermore, we assume that the difference between the output of the thinking model and the ground truth is controllable, i.e., $l(f(X_t), Y_t) \leq \delta$ for some small $\delta > 0$. Then by controlling the performance loss with respect to the thinking model $f(X_t)$ below $\epsilon$, we have $l(\hat{f}_t(X_t), Y_t) \leq \epsilon + \delta$, which means that the performance loss of B-PAC reasoning with respect to the ground truth is also controlled.

**Objective of efficient reasoning.** The primary objective of this work is to improve the *reasoning efficiency* of LRMs, not to improve the *capability* of thinking model. The premise of deploying a LRM is that users have already trusted its performance and are willing to pay for its superior reasoning capabilities. In this context, the goal of an efficient reasoning system is to approximate the behavior of the thinking model *as closely as possible* with minimal computational cost. Therefore, the performance loss in our setting should be defined by the fidelity gap between the composite model and the thinking model, rather than the inherent error of the thinking itself.

**Unavailability of ground truth in online settings.** In real-world online deployment such as open-ended question and code generation, the ground truth label $Y_t$ is typically unobservable, or requires prohibitively expensive human annotation. In contrast, the output of the thinking model $f(X_t)$ is *observable* if $\xi_t = 1$. These observable errors enable us to estimate the corresponding population risk (e.g., the proposed IPS estimator), and therefore controls its population level below the preset level with a high probability. If we aim to control the risk with respect to the ground truth, then we must have access to the ground truth, which is *impractical* for online generation tasks where immediate human oversight is absent.

## A.2. Impact of the Quality of Uncertainty Score

We provide a detailed discussion on the impact of uncertainty-score quality on the safety and efficiency of B-PAC reasoning.

**Safety of B-PAC reasoning.** Theorem 4.2 and Theorem 5.1 hold regardless of how $U_t$ is generated, therefore, the safety guarantees of B-PAC reasoning are entirely independent of the quality of the uncertainty scores. This means that even in high-stakes applications, users do not need to worry about the safety of using uncertainty score with potential low quality. Such flexibility enables users to choose different types of bounded uncertainty scores by their domain knowledge or prior information. For example, users can use logit-based scores derived from the token probabilities of the non-thinking model, or use the verbalized score generated by prompting the LLM to self-evaluate its confidence.

**Efficiency of B-PAC reasoning.** Although the safety of B-PAC reasoning is guaranteed regardless of uncertainty score, the reasoning efficiency is heavily dependent on the discriminative power of the uncertainty score. Generally speaking, when the uncertainty score can reflect, to some extent, the inconsistency between the output of the non-thinking model and the output of the thinking model (positive correlation), B-PAC reasoning will gain efficiency. The reasons are as follows.

The efficiency gain of B-PAC comes from its ability to identify the non-trivial safe threshold $\hat{u}_t > 0$. This requires the wealth $K_t(\hat{u}_t)$ to exceed a certain level. A high-quality $U_t$ ensures that the empirical risk is consistently lower than $\epsilon$ for a range of thresholds, allowing the gambler to win money and drive $\hat{u}_t$ upward. If the uncertainty score is of poor quality (e.g., uncorrelated with the actual error), the gambler will frequently lose games, causing the wealth $K_t(u)$ to shrink or stagnate.

To illustrate this clearly, we consider two extreme scenarios. Consider $\epsilon = 0.05$, $\rho_t = 0.1$, and the non-thinking model $\tilde{f}$ has a raw error rate of 0.2. Consider that $U_t$ perfectly distinguishes correct and incorrect outputs that $U_t = \mathbb{I}\{\tilde{f}(X_t) \neq f(X_t)\}$. In this scenario, for any threshold $u \in (0, 1)$, the set of accepted queries $\{X_t : U_t < u\}$ consists exclusively of correct instances. Consequently, the empirical risk within this accepted group is 0. Therefore, the gambler always wins the game $(D_t(u) > 0)$, causing the wealth $K_t(u)$ to grow exponentially for all $u$. As a result, B-PAC will identify and maintain a high threshold (e.g., $\hat{u}_t \approx 1$), routing nearly all easy queries to the non-thinking model and achieving maximum efficiency. In contrast, for the adversarial score $U_t = \mathbb{I}\{\tilde{f}(X_t) = f(X_t)\}$, the gambler will lose all money quickly.

### A.3. Engineering Considerations

In this subsection, we discuss the practical aspects of integrating B-PAC reasoning into real-world LLM serving systems.

**Negligible latency overhead.** Briefly, the total system-level computational cost of B-PAC for 1000 requests is only 0.046s. This is because updating does not require model retraining, gradient updates, or heavy matrix multiplications. We only maintain quantities over a discretized threshold grid, which just involves computing the IPS estimator and performing scalar multiplications per step. Compared to the substantial inference latency of LRMs that ranges from hundreds of milliseconds to seconds per query, the time cost of updating the threshold $\hat{u}_t$ is negligible.

**Asynchronous state management.** In high-concurrency scenarios, strict synchronization of the wealth process $K_t$ across all incoming requests may introduce lock contention. To address this, B-PAC reasoning can be implemented in an asynchronous manner. The routing decision can read the current threshold snapshot from a shared cache, while the wealth update runs in a background thread or a separate microservice upon receiving feedback.

**Scalability via distributed betting.** For large-scale services distributed across multiple clusters, maintaining a single global wealth process might be challenging. A practical engineering solution is to maintain sharded wealth processes, where different traffic segments (e.g., grouped by user tiers or domain topics) maintain their independent betting games. This not only solves the scalability bottleneck but also allows B-PAC to learn fine-grained routing policies tailored to specific data distributions (e.g., a "Math" betting process vs. a "Creative Writing" betting process), further enhancing overall efficiency.

## B. Technical Proofs

### B.1. Proof of Preliminary Lemma

To avoid ambiguity, let $\pi_t(u) = \mathbb{I}\{U_t \geq u\} + \rho_t \mathbb{I}\{U_t < u\}$ and $\xi_t(u) \sim \text{Bernoulli}(\pi_t(u))$.

**Lemma B.1.** *Let $r_t(u) = \mathbb{E}[l_t \mathbb{I}\{\xi_t(u) = 0\}|\mathcal{F}_{t-1}]$, and $D_t(u)$ defined by* (5). *We have*

$$r_t(u) = (1 - \rho_t)\mathbb{E}[l_t \mathbb{I}\{U_t < u\}|\mathcal{F}_{t-1}],$$

$$\mathbb{E}[D_t(u)|\mathcal{F}_{t-1}] = \epsilon - (1 - \rho_{min})\mathbb{E}[l_t \mathbb{I}\{U_t < u\}|\mathcal{F}_{t-1}] = \epsilon - \frac{1 - \rho_{min}}{1 - \rho_t} r_t(u).$$

*As a result, if $(X_t)_{t \geq 0}$ are i.i.d, we have $\mathbb{E}[D_t|\mathcal{F}_{t-1}] = \epsilon - \mathbb{E}[(1 - \rho_{min})l_t \mathbb{I}\{U_t < u\}]$ and $R_t(u) = (1 - \rho_t)\mathbb{E}[l_t \mathbb{I}\{U_t < u\}]$. Furthermore, for i.i.d. setting with $(\rho_t)_{t \geq 1}$ given by* (9), *we have*

$$R_t(u) = \frac{1 - \rho_{warm}}{1 - \rho_{deploy}} R(u)\mathbb{I}(t \leq T_{warm}) + R(u)\mathbb{I}(t > T_{warm}),$$

$$\mathbb{E}[D_t(u)|\mathcal{F}_{t-1}] = \epsilon - R(u).$$

*Proof of Lemma B.1.* For $D_t(u)$, we have

$$\mathbb{E}\left[\epsilon - Z_t(u)\Big|\mathcal{F}_{t-1}\right] = \mathbb{E}\left[\epsilon - \frac{(1-\rho_{\min})l_t\xi_t\mathbb{I}\{U_t < u\}}{\pi_t}\Big|\mathcal{F}_{t-1}\right]$$

$$\overset{(i)}{=} \epsilon - \mathbb{E}\left[\frac{(1-\rho_{\min})l_t\mathbb{I}\{U_t < u\}}{\pi_t}\mathbb{E}\left[\xi_t|\mathcal{F}_{t-1}, X_t\right]\Big|\mathcal{F}_{t-1}\right]$$

$$= \epsilon - \mathbb{E}\left[\frac{(1-\rho_{\min})l_t\mathbb{I}\{U_t < u\}}{\pi_t}\pi_t\Big|\mathcal{F}_{t-1}\right]$$

$$= \epsilon - (1-\rho_{\min})\mathbb{E}[l_t\mathbb{I}\{U_t < u\}|\mathcal{F}_{t-1}],$$

where $(i)$ holds since $\pi_t$ is a deterministic function given $\hat{u}_{t-1} \in \mathcal{F}_{t-1}$ and $X_t$. For $r_t(u)$, we have

$$r_t(u) = \mathbb{E}\left[l_t\mathbb{I}\{\xi_t(u) = 0\}|\mathcal{F}_{t-1}\right]$$

$$= \mathbb{E}[l_t\mathbb{I}\{U_t < u\}\mathbb{I}\{\xi_t(u) = 0\}|\mathcal{F}_{t-1}]$$

$$= \mathbb{E}\left[l_t\mathbb{I}\{U_t < u\}\mathbb{E}[\mathbb{I}\{\xi_t(u) = 0\}|\mathcal{F}_{t-1}, X_t]|\mathcal{F}_{t-1}\right]$$

$$= \mathbb{E}[l_t\mathbb{I}\{U_t < u\}(1 - \pi_t(u))|\mathcal{F}_{t-1}]$$

$$= \mathbb{E}[l_t\mathbb{I}\{U_t < u\}|\mathcal{F}_{t-1}] - \mathbb{E}[(\mathbb{I}\{U_t \geq u\} + \rho_t\mathbb{I}\{U_t < u\})l_t\mathbb{I}\{U_t < u\}|\mathcal{F}_{t-1}]$$

$$= \mathbb{E}[l_t\mathbb{I}\{U_t < u\}|\mathcal{F}_{t-1}] - \rho_t\mathbb{E}[l_t\mathbb{I}\{U_t < u\}|\mathcal{F}_{t-1}]$$

$$= (1 - \rho_t)\mathbb{E}[l_t\mathbb{I}\{U_t < u\}|\mathcal{F}_{t-1}].$$

Therefore, we also have

$$\mathbb{E}[D_t(u)|\mathcal{F}_{t-1}] = \epsilon - \frac{1 - \rho_{\min}}{1 - \rho_t}r_t(u).$$

Furthermore, if $(X_t)_{t\geq 0}$ are i.i.d., we have

$$\mathbb{E}[D_t(u)|\mathcal{F}_{t-1}] = \epsilon - (1-\rho_{\min})\mathbb{E}[l_t\mathbb{I}\{U_t < u\}|\mathcal{F}_{t-1}] = \epsilon - (1-\rho_{\min})\mathbb{E}[l_t\mathbb{I}\{U_t < u\}],$$

$$R_t(u) = \mathbb{E}[l_t\mathbb{I}\{\xi_t(u) = 0\}] = r_t(u) = (1-\rho_t)\mathbb{E}[l_t\mathbb{I}\{U_t < u\}|\mathcal{F}_{t-1}] = (1-\rho_t)\mathbb{E}[l_t\mathbb{I}\{U_t < u\}].$$

For $(\rho_t)_{t\geq 1}$ given by (9), we have $\rho_t = \rho_{\text{warm}}$ if $t \leq T_{\text{warm}}$ and $\rho_t = \rho_{\text{deploy}}$ if $t > T_{\text{warm}}$, and $\rho_{\min} = \rho_{\text{deploy}}$. This completes the proof. $\square$

**Lemma B.2** (Restatement of Lemma 4.1). *Let $\mathcal{F}_t$, $D_t(u)$, and $\rho_t$ be defined by (4), (5), and (9), respectively. Under the null hypothesis $H_{0,u} : R(u) > \epsilon$, for any nonnegative $\lambda_t(u) \in \mathcal{F}_{t-1}$ satisfying $1 + \lambda_t(u)D_t(u) \geq 0$, the process $(K_t(u))_{t\geq 0}$ (6) is a non-negative supermartingale with respect to $\mathcal{F}_t$.*

*Proof of Lemma B.2.*

$$\mathbb{E}[K_t(u) \mid \mathcal{F}_{t-1}] = \mathbb{E}\left[K_{t-1}(u) \cdot (1 + \lambda_t(u)D_t(u)) \mid \mathcal{F}_{t-1}\right]$$

$$\overset{(i)}{=} K_{t-1}(u)\left(1 + \lambda_t(u)\mathbb{E}[D_t(u) \mid \mathcal{F}_{t-1}]\right)$$

$$\overset{(ii)}{=} K_{t-1}(u)\left(1 + \lambda_t(u)\left(\epsilon - R(u)\right)\right)$$

$$\overset{(iii)}{\leq} K_{t-1}(u),$$

where $(i)$ holds since $\lambda_t(u)$ is predicable, $(ii)$ holds by Lemma B.1, and $(iii)$ holds by the null hypothesis $H_{0,u}$ that $R(u) > \epsilon$. This completes the proof. $\square$

**Lemma B.3** (Ville's inequality). *For the non-negative supermartingale $\{K_t(u)\}_{t=0}^{\infty}$ with $K_0 = 1$, for any $\alpha \in (0, 1)$, it holds that*

$$\mathbb{P}\left(\exists t : K_t(u) \geq \frac{1}{\alpha}\right) \leq \alpha.$$

**Lemma B.4.** *Consider real number $a_n \geq 0$, $1 \leq n \leq T$. Let $S_t = \sum_{i=1}^{t} a_i + \beta$, where $\beta > 0$, $1 \leq t \leq T$. Set $S_0 = \beta$. If $a_t \leq \gamma S_{t-1}$ holds for all $1 \leq t \leq T$, then*

$$\sum_{t=1}^{T} \frac{a_t}{S_{t-1}} \leq C(\gamma) \log\left(\frac{S_T}{\beta}\right),$$

*where $C(\gamma) = \gamma/\log(1+\gamma)$.*

*Proof of Lemma B.4.* Note that if there exists a constant $C$ such that $a_t/S_{t-1} \leq C(\log S_t - \log S_{t-1})$ holds for all $t$, then

$$\sum_{t=1}^{T} \frac{a_t}{S_{t-1}} \leq C(\log S_T - \log S_0) = C \log\left(\frac{S_T}{\beta}\right).$$

For $t \geq 1$,

$$\log S_t - \log S_{t-1} = \log\left(\frac{S_t}{S_{t-1}}\right) = \log\left(1 + \frac{a_t}{S_{t-1}}\right).$$

Let $x_t = a_t/S_{t-1} \leq \gamma$. It remains to prove $x_t \leq C\log(1+x_t)$ for $x_t \in [0,\gamma]$ and $C = \gamma/\log(1+\gamma)$. It can be verified that

$$\max_{0 < x \leq \gamma} \frac{x}{\log(1+x)} = \frac{\gamma}{\log(1+\gamma)},$$

which completes the proof. $\square$

### B.2. Proof of Theorem 4.2

*Proof of Theorem 4.2.* Define the oracle threshold by

$$u^* := \min\left\{u \in \mathcal{U} : R(u) > \epsilon\right\}.$$

Note that the deployment risk $R(u) = (1 - \rho_{\min})\mathbb{E}[l_t \mathbb{I}\{U_t < u\}]$ is non-decreasing, i.e.,

$$R(u) \leq R(u') \quad \text{for any } u \leq u'.$$

Therefore, we have

$$\{\exists t \geq 1, \text{ s.t. } R(\hat{u}_t) > \epsilon\} = \{\exists t \geq 1, \text{ s.t. } \hat{u}_t \geq u^*\}. \tag{14}$$

By the definition of $\hat{u}_t$ that

$$\hat{u}_t = \max\left\{u^{(i)} \in \mathcal{U} : \forall j \leq i, K_t(u^{(j)}) \geq \frac{1}{\alpha}\right\},$$

we have $K_t(\hat{u}_t) \geq 1/\alpha$. Again, by the fixed sequence testing, we have

$$\{\exists t \geq 1, \text{ s.t. } \hat{u}_t \geq u^*\} \subseteq \left\{\exists t \geq 1, \text{ s.t. } K_t(u^*) \geq \frac{1}{\alpha}\right\}. \tag{15}$$

By the definition of $u^*$, the null hypothesis $H_{0,u^*} : R(u^*) > \epsilon$ holds. Therefore, by Lemma B.2, $(K_t(u^*))_{t \geq 0}$ is a non-negative supermartingale. By Lemma B.3, we have

$$\mathbb{P}\left(\exists t \geq 1, K_t(u^*) \geq \frac{1}{\alpha}\right) \leq \alpha. \tag{16}$$

According to (14), (15), and (16), we have

$$\mathbb{P}\left(\exists t \geq 1, R(\hat{u}_t) > \epsilon\right) \leq \mathbb{P}\left(\exists t \geq 1, K_t(u^*) \geq \frac{1}{\alpha}\right) \leq \alpha.$$

Therefore, we have

$$\mathbb{P}(\forall t : R(\hat{u}_t) \leq \epsilon) \geq 1 - \alpha, \tag{17}$$

By Lemma B.1, for $\rho_{\text{warm}} \geq \rho_{\text{deploy}}$, we have

$$R_t(u) = \frac{1 - \rho_{\text{warm}}}{1 - \rho_{\text{deploy}}} R(u)\mathbb{I}(t \leq T_{\text{warm}}) + R(u)\mathbb{I}(t > T_{\text{warm}}) \leq R(u).$$

Therefore, we have

$$\mathbb{P}(\forall t : R_t(\hat{u}_t) \leq \epsilon) \geq \mathbb{P}(\forall t : R(\hat{u}_t) \leq \epsilon) \geq 1 - \alpha, \tag{18}$$

which completes the proof. □

### B.3. Proof of Theorem 4.3

*Proof of Theorem 4.3.* Fixed $u \in \mathcal{U}$. Since $l_t \in [0, 1]$ and the exploration probability is bounded below by $\rho_t > 0$, we have $D_t(u) \in [\epsilon - (1 - \rho_{\text{deploy}})/\rho_t, \epsilon]$. Let $M_t = \max\{\epsilon, (1 - \rho_{\text{deploy}})/\rho_t - \epsilon\}$. Denote the feasible set of betting fractions by $\mathcal{K}_t = [0, c/M_t]$ for some constant $c \in (0, 1)$. Therefore, for any fixed $\lambda \in \mathcal{K}_t$ and any $D_t(u)$, we have

$$1 + \lambda D_t(u) \geq 1 - c > 0. \tag{19}$$

We define the surrogate objective $\Phi_{t-1}(\lambda)$ as the sum of quadratic lower bounds plus a regularizer:

$$\Phi_{t-1}(\lambda) = \sum_{i=1}^{t-1}(\lambda D_i(u) - \frac{1}{2}\lambda^2 D_i^2(u)) - \frac{\beta}{2}\lambda^2.$$

We first prove that the $\lambda_t$ given by (8) satisfies

$$\lambda_t(u) = \arg\max_{\lambda \in \mathcal{K}_t} \Phi_{t-1}(\lambda),$$

with $\beta = 1$. The gradient of $\Phi_{t-1}(\lambda)$ with respect to $\lambda$ is

$$\nabla_\lambda \Phi_{t-1}(\lambda) = \sum_{i=1}^{t-1} D_i(u) - \lambda \sum_{i=1}^{t-1} D_i^2(u) - \beta\lambda.$$

By setting $\nabla_\lambda \Phi_{t-1}(\lambda) = 0$, the unconstrained solution of $\lambda$ is

$$\lambda_t^{\text{raw}}(u) := \frac{\sum_{i=1}^{t-1} D_i(u)}{\sum_{i=1}^{t-1} D_i^2(u) + \beta}.$$

By straightforward computation, we can check that $\Phi_{t-1}(\lambda)$ is a concave function of $\lambda$. Since the constraint set $\mathcal{K}_t$ is convex, the constrained optimum is simply the projection of $\lambda_t^{\text{raw}}(u)$ onto $\mathcal{K}_t$:

$$\arg\max_{\lambda \in \mathcal{K}_t} \Phi_{t-1}(\lambda) = \text{proj}_{\mathcal{K}_t}(\lambda_t^{\text{raw}}(u)) = \min\left\{\max\left\{\frac{\sum_{i=0}^{t-1} D_i(u)}{\sum_{i=0}^{t-1} D_i^2(u) + \beta}, 0\right\}, \frac{c}{M_t}\right\}.$$

We complete the proof by choosing $\beta = 1$.

Second, we derive the regret bound as follows. Let $g_t(\lambda) = \lambda D_t(u) - \lambda^2 D_t^2(u)/2$ be the quadratic proxy. The regularizer is given by $\psi(\lambda) = \beta\lambda^2/2$. By the standard FTRL bound on $g_t$ (Hazan, 2016), we have

$$\sum_{t=1}^{T}(g_t(\lambda^*) - g_t(\lambda_t)) \leq \psi(\lambda^*) - \psi(\lambda_1) + \frac{1}{2}\sum_{t=1}^{T} \frac{(\nabla g_t(\lambda_t))^2}{-\nabla^2 \Phi_{t-1}(\lambda_t)}. \tag{20}$$

Since $\lambda_T^* \in [0, c/M_T]$, we have

$$\psi(\lambda_T^*) - \psi(\lambda_1) \leq \frac{\beta}{2}(\lambda_T^*)^2 \leq \frac{\beta c^2}{2M_T^2}. \tag{21}$$

For $\nabla^2 \Phi_{t-1}(\lambda_t)$, we have $-\nabla^2 \Phi_{t-1}(\lambda_t) = \sum_{i=1}^{t-1} D_i^2(u) + \beta$. For $\nabla g_t(\lambda_t)$, we have

$$(\nabla g_t(\lambda_t))^2 = \left(D_t(u) - \lambda_t D_t^2(u)\right)^2 = D_t^2(u)\left(1 - \lambda_t D_t(u)\right)^2.$$

By $|\lambda_t D_t(u)| \le c$, we have

$$(\nabla g_t(\lambda_t))^2 \le (1+c)^2 D_t^2(u). \tag{22}$$

By (22), we have

$$\sum_{t=1}^{T} \frac{(\nabla g_t(\lambda_t))^2}{-\nabla^2 \Phi_{t-1}(\lambda_t)} \le \sum_{t=1}^{T} \frac{(1+c)^2 D_t^2(u)}{\sum_{i=1}^{t-1} D_i^2(u) + \beta} = (1+c)^2 \sum_{t=1}^{T} \frac{D_t^2(u)}{\sum_{i=1}^{t-1} D_i^2(u) + \beta}. \tag{23}$$

Note that

$$\frac{D_t^2(u)}{\sum_{i=1}^{t-1} D_i^2(u) + \beta} \le \frac{M_t^2}{\beta}.$$

Let $M = \sup_{t \ge 1} M_t$. By Lemma B.4 , we have

$$\sum_{t=1}^{T} \frac{D_t^2(u)}{\sum_{i=1}^{t-1} D_i^2(u) + \beta} \le \frac{M^2/\beta}{\log(1 + M^2/\beta)} \log\left( \frac{\sum_{t=1}^{T} D_t^2(u)}{\beta} + 1 \right). \tag{24}$$

Combining (20), (21), (23), and (24), we have

$$\mathcal{R}_T^{\text{quad}} \le \frac{\beta c^2}{2M_T^2} + \frac{(1+c)^2 M^2}{2\beta \log(1 + M^2/\beta)} \log\left( \frac{\sum_{t=1}^{T} D_t^2(u)}{\beta} + 1 \right).$$

For $T > T_{\text{warm}}$, we have $M_T = M$. Using the worst-case bound $\sum D_t^2(u) \le TM^2$, we have

$$\mathcal{R}_T^{\text{quad}} \le \frac{\beta c^2}{2M^2} + \frac{(1+c)^2 M^2}{2\beta \log(1 + M^2/\beta)} \log\left( \frac{TM^2}{\beta} + 1 \right).$$

By choosing $\beta = 1$, we complete the proof of Theorem 4.3. $\qquad\square$

## B.4. Proof of Theorem 5.1

*Proof of Theorem 5.1.* Let

$$Y_t(u) = \prod_{i=1}^{t} \frac{1 + \lambda_i(u) D_i(u)}{1 + \lambda_i(u) \mathbb{E}[D_i(u)|\mathcal{F}_{i-1}]}, \quad H_t = \prod_{i=1}^{t} (1 + \lambda_i(u) \mathbb{E}[D_i(u)|\mathcal{F}_{i-1}]).$$

We have $K_t(u) = Y_t(u) H_t(u)$. Note that $(Y_t)_{t \ge 0}$ is a $(\mathcal{F}_t)$-adapted martingale since

$$\mathbb{E}[Y_t(u)|\mathcal{F}_{t-1}] = Y_{t-1} \mathbb{E}\left[ \frac{1 + \lambda_t(u) D_t(u)}{1 + \lambda_t(u) \mathbb{E}[D_t(u)|\mathcal{F}_{t-1}]} \Big| \mathcal{F}_{t-1} \right] = Y_{t-1}.$$

It holds that $\mathbb{E}[Y_t] = 1$ for $t \ge 0$. In addition, by $1 + x \le e^x$, we have

$$
\begin{aligned}
H_t(u) &\le \prod_{i=1}^{t} \exp\left\{ \lambda_i(u) \mathbb{E}[D_i(u)|\mathcal{F}_{i-1}] \right\} \\
&= \exp\left\{ \sum_{i=1}^{t} \lambda_i(u) \mathbb{E}[D_i(u)|\mathcal{F}_{i-1}] \right\} \\
&\overset{(i)}{=} \exp\left\{ \sum_{i=1}^{t} \lambda_i(u) (\epsilon - r_i(u)) \right\} \\
&= \exp\left\{ \sum_{i=1}^{t} \lambda_i(u) \left( \epsilon - \frac{\sum_{j=1}^{t} \lambda_j(u) r_j(u)}{\sum_{i=1}^{t} \lambda_i(u)} \right) \right\} \\
&\overset{(ii)}{=} \exp\left\{ \sum_{i=1}^{t} \lambda_i(u) (\epsilon - L_t(u)) \right\},
\end{aligned}
\tag{25}
$$

where $(i)$ and $(ii)$ hold by Lemma B.1 and (13).

Let $M_t = \sum_{u \in \mathcal{U}} w(u) Y_t(u)$. We have

$$\mathbb{E}[M_t | \mathcal{F}_{t-1}] = \sum_{u \in \mathcal{U}} w(u) \mathbb{E}[Y_t(u) | \mathcal{F}_{t-1}] = \sum_{u \in \mathcal{U}} w(u) Y_{t-1}(u) = M_{t-1},$$

$$\mathbb{E}[M_t] = \sum_{u \in \mathcal{U}} w(u) \mathbb{E}[Y_t(u)] = 1.$$

Therefore, the process $(M_t)_{t \geq 0}$ is also a $(\mathcal{F}_t)_{t \geq 0}$-adapted martingale. Therefore, by Lemma B.3, we have

$$\mathbb{P}\left( \exists t \in \mathbb{N} : M_t \geq \frac{1}{\alpha} \right) \leq \alpha. \tag{26}$$

Let

$$\mathcal{E} = \left\{ \exists t \in \mathbb{N}, \exists u^* \in \mathcal{U} : L_t(u^*) > \epsilon, K_t(u^*) \geq \frac{1}{\alpha \nu(u^*)} \right\}.$$

On the event $\mathcal{E}$, by (25), we have

$$\frac{1}{\alpha \nu(u^*)} \leq K_t(u^*) = Y_t(u^*) H_t(u^*) \leq Y_t(u^*). \tag{27}$$

By (27), we have

$$M_t = \sum_{u \in \mathcal{U}} \nu(u) Y_t(u) \geq \nu(u^*) Y_t(u^*) \geq \nu(u^*) \frac{1}{\alpha \nu(u^*)} = \frac{1}{\alpha}.$$

By (26) and (27), we have

$$\mathbb{P}(\mathcal{E}) \leq \mathbb{P}\left( \exists t \in \mathbb{N} : M_t \geq \frac{1}{\alpha} \right) \leq \alpha.$$

Let $\hat{\mathcal{S}}_t = \{ u \in \mathcal{U} : K_t(u) \geq 1/(\alpha \nu(u)) \}$. Recall that $\hat{u}_t$ is the element of $\hat{\mathcal{S}}_t$ with largest value. We have

$$\mathbb{P}(\exists t \in \mathbb{N} : L_t(\hat{u}_t) > \epsilon) \leq \mathbb{P}(\exists t \in \mathbb{N} : \exists u^* \in \hat{\mathcal{S}}_t, L_t(u^*) > \epsilon) \leq \mathbb{P}(\mathcal{E}) \leq \alpha.$$

Therefore, we have

$$\mathbb{P}(\forall t \in \mathbb{N} : L_t(\hat{u}_t) \leq \epsilon) \geq 1 - \alpha.$$

This completes the proof. □

## C. Experimental Details

### C.1. Details of Datasets

Given the substantial computational cost of evaluating long reasoning chains, we randomly sampled an initial pool of instances: 6,000 for Magpie, 5,000 for MATH, 4,992 for MMLU-Pro, and 6,511 for BBH. Following the protocol in (Zeng et al., 2025) and our discussion in Appendix A, we focus on scenarios where the thinking model demonstrates superior or the same capability over the non-thinking counterpart. Specifically, for reasoning tasks (MATH, MMLU-Pro, BBH), we retained only instances where the thinking model predicted the correct answer. For the open-ended Magpie dataset, we selected instances where the thinking model achieved a higher preference score. This filtering process yielded the final datasets for sequential evaluation: MMLU-Pro (3,738), MATH (4,628), BBH (5,476), and Magpie (4,379). Furthermore, we constructed two specialized evaluation streams. First, to demonstrate the efficiency gain from dynamic threshold relaxation (Figure 1), we created a mixed dataset comprising all valid Magpie samples and a random subset of 100 BBH samples. Second, to evaluate robustness against non-stationary data (Figure 2), we synthesized a distribution-shift stream by combining 1,500 samples from MMLU-Pro and 3,000 samples from BBH.

## C.2. Detailed Settings of LLMs

In this study, we adhere to specific decoding configurations for different model variants. For Qwen3-4B-Instruct-2507, we set the hyperparameters as follows: temperature $T = 0.7$, Top-$P = 0.8$, Top-$K = 20$, and min-$P = 0$. For the reasoning-focused Qwen3-4B-Thinking-2507, we adopt a slightly more deterministic setting with $T = 0.6$, Top-$P = 0.95$, Top-$K = 20$, and min-$P = 0$. All experiments were conducted on NVIDIA A100-SXM4 GPU (40GB). For the evaluation of the Magpie dataset, we employ GPT-4o-mini as the judge via the official API, utilizing default parameter settings. The corresponding prompt is presented at Figure 4.

---

**Evaluation Prompt**

You are a careful, unbiased and strict evaluator.

Given a user question and an assistant answer, evaluate the overall quality of the answer. Consider the following aspects:
- Correctness and factual accuracy
- Helpfulness and relevance to the question
- Clarity and completeness
- Following the user's instructions

Provide a single numeric score from 1 to 10, where:
1 = very poor answer
10 = excellent answer

Do not provide any explanation. Output only the number.

User Question:
{question}

Assistant Answer:
{answer}

---

*Figure 4.* Prompt used for evaluating response quality on Magpie dataset.

## C.3. Details of Baselines and B-PAC reasoning

In this subsection, we introduce two baselines for online efficient reasoning, including O-naive and IPS+Hoeff. Notations are consistent with those of B-PAC reasoning. Then we present the settings of B-PAC reasoning in our experiments.

**O-Naive.** This baseline represents a heuristic strategy adopted in scenarios with partial feedback. It operates under the optimistic assumption that unobserved instances are error-free. Specifically, if the thinking model is not invoked ($\xi_t = 0$), then the unobserved loss is regarded as $0$. Consequently, the risk is estimated by the average of the observed losses:

$$\hat{R}_t^{\text{o-naive}}(u) = \frac{1}{t} \sum_{i=1}^{t} (\xi_i l_i + (1 - \xi_i) \cdot 0) \cdot \mathbb{I}(U_i < u) = \frac{1}{t} \sum_{i=1}^{t} \xi_i l_i \mathbb{I}(U_i < u).$$

Then O-Naive greedily selects the maximum threshold $\hat{u}_t$ satisfying $\hat{R}_t^{\text{o-naive}}(\hat{u}_t) \leq \epsilon$, i.e.,

$$\hat{u}_t = \max\{u \in \mathcal{U} : \hat{R}_t^{\text{o-naive}}(u) \leq \epsilon\}.$$

**IPS+Hoeff.** We provide another anytime safe method for efficient reasoning, named IPS+Hoeff. Unlike O-Naive, this baseline leverages the IPS estimator and the Hoeffding's inequality combined with a union bound to ensure the anytime safety. Specifically, let $Z_t(u) = (1 - \rho)l_t\xi_t\mathbb{I}\{U_t < u\}/\pi_t$ as in (3). Let $\tilde{M} = (1 - \rho)/\rho$ and $\alpha_t = 6\alpha/(\pi^2 t^2)$. At time $t$, IPS+Hoeff selects the maximum threshold $\hat{u}_t$ such that the upper confidence bound (UCB) of the risk is within $\epsilon$, i.e.,

$$\hat{u}_t = \max\left\{u \in \mathcal{U} : \frac{1}{t} \sum_{i=1}^{t} Z_i(u) + \tilde{M}\sqrt{\frac{\log(N/\alpha_t)}{2t}} \leq \epsilon\right\}.$$

**Proposition C.1.** *Suppose $(X_t)_{t \geq 1}$ are independent and identically distributed. The method of IPS+Hoeff satisfies that* $\mathbb{P}(\exists t \geq 1 : R_t(\hat{u}_t) > \epsilon) \leq \alpha$.

*Proof.* For any fixed threshold $u \in \mathcal{U}$. The IPS estimator $Z_1(u), \ldots, Z_t(u)$ are independent and bounded in $[0, \tilde{M}]$. By Hoeffding's inequality, we have

$$\mathbb{P}\left(R_t(u) > \frac{1}{t}\sum_{i=1}^t Z_i(u) + \eta\right) \leq \exp\left(-\frac{2t\eta^2}{\tilde{M}^2}\right).$$

Let $\exp\left(-2t\eta^2/\tilde{M}^2\right) = \alpha_t/N$. We have $\eta = \tilde{M}\sqrt{\log(N/\alpha_t)/2t}$. Then we have

$$\mathbb{P}\left(\exists u \in \mathcal{U} : R_t(u) > \frac{1}{t}\sum_{i=1}^t Z_i(u) + \tilde{M}\sqrt{\frac{\log(N/\alpha_t)}{2t}}\right) \leq \sum_{u\in\mathcal{U}}\frac{\alpha_t}{K} = \alpha_t.$$

Finally, applying the union bound over $t \geq 1$, we have

$$\mathbb{P}\left(\exists t : \exists u \in \mathcal{U} : R_t(u) > \frac{1}{t}\sum_{i=1}^t Z_i(u) + \tilde{M}\sqrt{\frac{\log(N/\alpha_t)}{2t}}\right) \leq \sum_{t=1}^\infty \alpha_t = \sum_{t=1}^\infty \frac{6\alpha}{\pi^2 t^2} = \alpha.$$

This completes the proof. $\qquad\square$

We note that a stronger baseline with a tighter UCB may be derived by leveraging the idea of bootstrap or random weighting (Ming et al., 2028). But, in our experiment, we choose $\hat{u}_t$ by

$$\hat{u}_t = \max\left\{u \in \mathcal{U} : \frac{1}{t}\sum_{i=1}^t Z_i(u) + \tilde{M}\sqrt{\frac{\log(1/\alpha_t)}{2t}} \leq \epsilon\right\},$$

which is more efficient than the initial IPS+Hoeff. Under this strategy, IPS+Hoeff still yields no efficiency, which demonstrates the necessity of the proposed betting framework.

**Settings of B-PAC reasoning.** We take $(\rho_t)_{t\geq1}$ and $(\lambda_t(u))_{t\geq1}$ as in (9) and (8), respectively. Specifically, we choose $T_{\text{warm}} = 200$, $\rho_{\text{warm}} = 0.7$, and $\rho_{\text{deploy}} = 0.05$ for $(\rho_t)_{t\geq1}$. In this case, we have $\rho_{\min} = 0.05$. For (8), we simply choose $c = 0.9$. The search space of threshold is $\mathcal{U} = \{0, 0.001, 0.002, \ldots, 1\}$ and the threshold $\hat{u}_t$ is determined by (7). Note that for all experiments, we adopt the above settings without making any adjustments involving hyperparameters.

### C.4. Details of Uncertainty Scores

Across this paper, we use the logits score (Zeng et al., 2025; Hao et al., 2023; Huang et al., 2025) as uncertainty score for both B-PAC reasoning and the method of Zeng et al. (2025). Specifically, let $\tilde{f}(X_t) = \tilde{y}_t = (\tilde{y}_{t,1}, \ldots, \tilde{y}_{t,\tilde{h}(X_i)})$ denote the output of non-thinking model $\tilde{f}(X_t)$ consisting of $\tilde{h}(X_i)$ tokens, where $\tilde{y}_{t,j}$ representing the $j$-th token of $\tilde{y}_t$. The uncertainty score of $\tilde{f}(X_t)$ is defined by the averaged token probability:

$$U_t = 1 - \frac{1}{\tilde{h}(X_t)}\sum_{j=1}^{\tilde{h}(X_t)}\mathbb{P}(\tilde{y}_{t,j}|y_{t,1}, \ldots, y_{t,j-1}, X_t),$$

where $\mathbb{P}(\tilde{y}_{t,j}|y_{t,1}, \ldots, y_{t,j-1}, X_t)$ is the conditional probability of token $\tilde{y}_{t,j}$ computed from the prediction logits.

### C.5. Details of Loss Functions

**Binary 0-1 loss.** Let $Y_t$ be the ground truth of question $X_t$. For scenarios where the correctness of the answer is verifiable such as in mathematical problem-solving or multiple-choice question answering, we use the binary 0–1 loss:

$$l(\hat{f}_t(X_t), f(X_t)) = \mathbb{I}\{\hat{f}_t(X_t) \neq f(X_t)\}\mathbb{I}\{f(X_t) = Y_t\}.$$

**LLM-judge loss.** For open-ended tasks, exact matching is inapplicable. We employ an LLM-as-a-Judge to score responses. Specifically, let $s(\cdot)$ be the score assigned by an LLM judge. We define the loss as the normalized square root of the score by

$$l(\hat{f}_t(X_t), f(X_t)) = \sqrt{\frac{\max\{0, s(f(X_t)) - s(\hat{f}_t(X_t))\}}{\|s\|}},$$

where $\|s\|$ denotes the maximum possible score range.

**Multi-type task loss.** We define a task-dependent bounded loss function as follows. Let $\mathcal{D}_{\text{verifiable}}$ denote tasks with ground truth (e.g., MATH), and let $\mathcal{D}_{\text{open-ended}}$ denote open-ended generation tasks. Depending on the task type, the loss is formulated by

$$l(\hat{f}_t(X_t), f(X_t)) = \begin{cases} \mathbb{I}\{\hat{f}_t(X_t) \neq f(X_t)\}, & \text{if } X_t \in \mathcal{D}_{\text{verifiable}}, \\ \sqrt{\frac{\max\{0, s(f(X_t)) - s(\hat{f}_t(X_t))\}}{\|s\|}}, & \text{if } X_t \in \mathcal{D}_{\text{open-ended}}. \end{cases}$$

*Remark* C.2. As discussed in Appendix A, it is reasonable to only consider data satisfying $f(X_t) = Y_t$ for verifiable tasks and $f(X_t) \geq \tilde{f}(X_t)$ for open-ended tasks. In this case, we have $\mathbb{I}\{f(X_t) = Y_t\} = 1$ for verifiable tasks and $\max\{0, s(f(X_t)) - s(\hat{f}_t(X_t))\} = s(f(X_t)) - s(\hat{f}_t(X_t))$ for open-ended tasks.

*Remark* C.3. We note that the LLM-judge loss may have inherent randomness, i.e., $l(\hat{f}_t(X_t), f(X_t))$ is a random variable given $\hat{f}_t(X_t)$ and $f(X_t)$. However, this randomness usually does not compromise the safety guarantee. Specifically, let $\tilde{l}(\hat{f}_t(X_t), f(X_t); e_t)$ be the random loss, where noises $(e_t)_{t \geq 1}$ are i.i.d. and independent of $(X_t)_{t \geq 1}$. By the same proof of Theorem 4.2, we have

$$\mathbb{P}(\forall t \geq 1 : V_t(\hat{f}_t) \leq \epsilon) \geq 1 - \alpha,$$

where $V_t(\hat{f}_t) = \mathbb{E}_{X \sim P_X, e_t \sim e}[\tilde{l}(\hat{f}_t(X_t), f(X_t); e_t)]$. In the common additive-noise model where

$$\tilde{l}(\hat{f}_t(X_t), f(X_t); e_t) = l(\hat{f}_t(X_t), f(X_t)) + e_t,$$

we have $V_t(\hat{f}_t) = R_t(\hat{f}_t)$ by $\mathbb{E}[e_t] = 0$. Therefore, safety remains; the inherent randomness only introduces extra variance and may reduce efficiency.

*Remark* C.4. B-PAC reasoning only requires the loss function to be bounded, allowing us to use a variety of loss functions according to domain knowledge. For example, for open-ended questions, one can also use the semantic cosine distance in an embedding space (Zeng et al., 2025), i.e.,

$$l(\hat{f}_t(X_t), f(X_t)) = 1 - \frac{v_{\hat{f}(X_t)} \cdot v_{f(X_t)}}{\|v_{\hat{f}(X_t)}\|\|v_{f(X_t)}\|},$$

where $v_{\hat{f}(X_t)}$ and $v_{f(X_t)}$ denote the embedding of $\hat{f}_t(X_t)$ and $f(X_t)$, respectively.

# D. Additional Experimental Results

## D.1. Performance under Other Uncertainty Scores

We also evaluate the performance of B-PAC reasoning with uncertainty scores including Perplexity (PPL) and Entropy. The settings for B-PAC reasoning are identical as in the former experiments.

*Table 2.* Results with different uncertainty scores on Magpie and MMLU-Pro. All values are reported as mean ± standard deviation.

| Dataset | Uncertainty Score | ER | ECP(%) | TP(%) |
|---------|-------------------|-----|--------|-------|
| Magpie | PPL | $0.0551 \pm 0.007$ | $22.76 \pm 7.57$ | $61.69 \pm 7.22$ |
| Magpie | Entropy | $0.0552 \pm 0.006$ | $23.12 \pm 6.51$ | $61.84 \pm 6.18$ |
| MMLU-Pro | PPL | $0.0319 \pm 0.011$ | $43.24 \pm 8.19$ | $75.12 \pm 9.26$ |
| MMLU-Pro | Entropy | $0.0316 \pm 0.012$ | $43.08 \pm 8.66$ | $74.59 \pm 9.83$ |

By Table 2, B-PAC remains safe (ER<0.08) and efficient (small ECP and TP) for different uncertainty scores.

## D.2. Performance under Different Tolerance Level

We investigate the impact of the user-specified error tolerance $\epsilon$ on the performance of B-PAC reasoning. We evaluate B-PAC on the MMLU-Pro, BBH, MATH, and Magpie with $\epsilon$ ranging from 0.05 to 0.10. As illustrated in Figure 5, B-PAC consistently maintains the empirical risk below the target $\epsilon$ across all levels. Furthermore, the results exhibit a clear trade-off between safety and efficiency: as $\epsilon$ decreases (imposing stricter safety constraints), B-PAC adaptively increases the expert call percentage (ECP) to ensure reliability, leading to lower token savings. Conversely, a looser tolerance allows the framework to route more queries to the non-thinking model, thereby improving efficiency.

### D.3. Further Comparison with PAC Reasoning

As shown in Figure 1 and 2, B-PAC reasoning can enjoy higher efficiency and rigorous safety than PAC reasoning (Zeng et al., 2025) for non-stationary settings. However, it is important to acknowledge that B-PAC reasoning guarantees anytime-valid performance loss control, a significantly stricter constraint than the fixed-sample validity of standard PAC reasoning (Zeng et al., 2025). Theoretically, this rigorous safety comes at a cost, which necessitates a more conservative strategy (lower efficiency) in the early stages. In this subsection, we demonstrate that this trade-off is transient. As the time step $t$ increases and evidence accumulates, the betting martingale tightens its bounds, allowing the efficiency of B-PAC to gradually converge to, and potentially surpass, that of the offline PAC baseline.

To demonstrate this, we present the comparison results on Magpie and MATH in Figure 6. On Magpie, the dynamic threshold $\hat{u}_t$ of B-PAC reasoning eventually surpasses the fixed threshold of PAC reasoning, potentially resulting in superior long-term efficiency (lower ECP and TP). On MATH, $\hat{u}_t$ converges closely to the PAC baseline. We attribute the small gap in ECP and TP to the mandatory exploration overhead. Specifically, unlike offline PAC, B-PAC maintains a minimum exploration probability ($\rho_{\text{deploy}} = 0.05$) to ensure unbiased risk estimation via IPS. This cost is the necessary premium for maintaining anytime safety and the ability to adapt to potential future shifts.

### D.4. Ablation Study on Impact of the Adaptive Betting Strategy

We investigate the contribution of the adaptive betting strategy $\lambda_t(u)$ given by (8) to the efficiency of B-PAC. Under the two-stage strategy $\rho_t$, to guarantee $1 + \lambda_t D_t \geq 0$, the fixed betting fraction $\lambda_t$ should satisfy $\lambda_t \leq 1/(1/\rho_{\min} - 1 - \epsilon) \approx 0.053$. Therefore, we choose $\lambda_t = 0.05$ for the fixed strategy. Figure 7 presents the performance of B-PAC using our proposed adaptive method against a baseline with a fixed $\lambda_t = 0.05$. While both methods achieve anytime-valid performance loss control (ER $< \epsilon$), the adaptive strategy demonstrates significantly faster convergence in terms of efficiency. As shown in the ECP and TP curves, the adaptive approach rapidly reduces expert invocations in the early stages, whereas the fixed $\lambda_t$ approach remains overly conservative, yielding a much slower decline in computational cost. This disparity arises because the fixed $\lambda_t$ fails to capitalize on the accumulated wealth. In contrast, our adaptive strategy dynamically scales the wager size based on past evidence, allowing the framework to accelerate threshold updates when the empirical risk is low, thereby achieving a superior trade-off between safety and efficiency.

### D.5. Ablation Study on Impact of the Exploration Strategy

**Fixed $\rho_t$ vs two-stage $\rho_t$.** We conduct an ablation study to validate the design of our two-stage exploration strategy. We compare the two-stage strategy against baselines with fixed exploration probabilities $\rho_t \in \{0.05, 0.2, 0.3, 0.7\}$ on MMLU-Pro and MATH; see Figure 8. Across the two benchmarks, our two-stage strategy consistently achieves the highest efficiency, revealing the inherent limitation of fixed exploration strategies. Specifically, a small exploration probability (e.g., $\rho_t \equiv 0.05$) causes the wealth process to accumulate positive evidence slowly. Consequently, the routing threshold $\hat{u}_t$ remains conservative for a prolonged period, resulting in unnecessarily high expert invocations during the early stages. In contrast, a large $\rho_t$ facilitates rapid threshold updates initially but incurs a high exploration cost. When a near-optimal safe threshold is identified, the system is still forced to query the thinking model for at least $\rho_t \times 100\%$ of instances. As discussed in Subsection 3.3, our two-stage strategy effectively searches for the optimal threshold in the early stage and reduces exploration cost in the later stage.

**Sensitivity to warm-up duration $T_{\text{warm}}$.** We further investigate the robustness of B-PAC reasoning with respect to the warm-up duration $T_{\text{warm}}$. Figure 9 compares the performance of B-PAC reasoning across $T_{\text{warm}} \in \{10, 50, 100, 200, 300, 500\}$ on Magpie and MATH. As discussed above and reflected by Figure 9, $T_{\text{warm}} = 10$ is not efficient since the two-stage strategy degrades to the fixed strategy if $T_{\text{warm}} \to 0$. Conversely, an excessively large $T_{\text{warm}}$ prolongs the high-exploration phase, incurring unnecessary computational overhead for exploration. Crucially, B-PAC reasoning demonstrates strong robustness across a wide range of intermediate values, suggesting that precise hyperparameter tuning is not required for deployment.

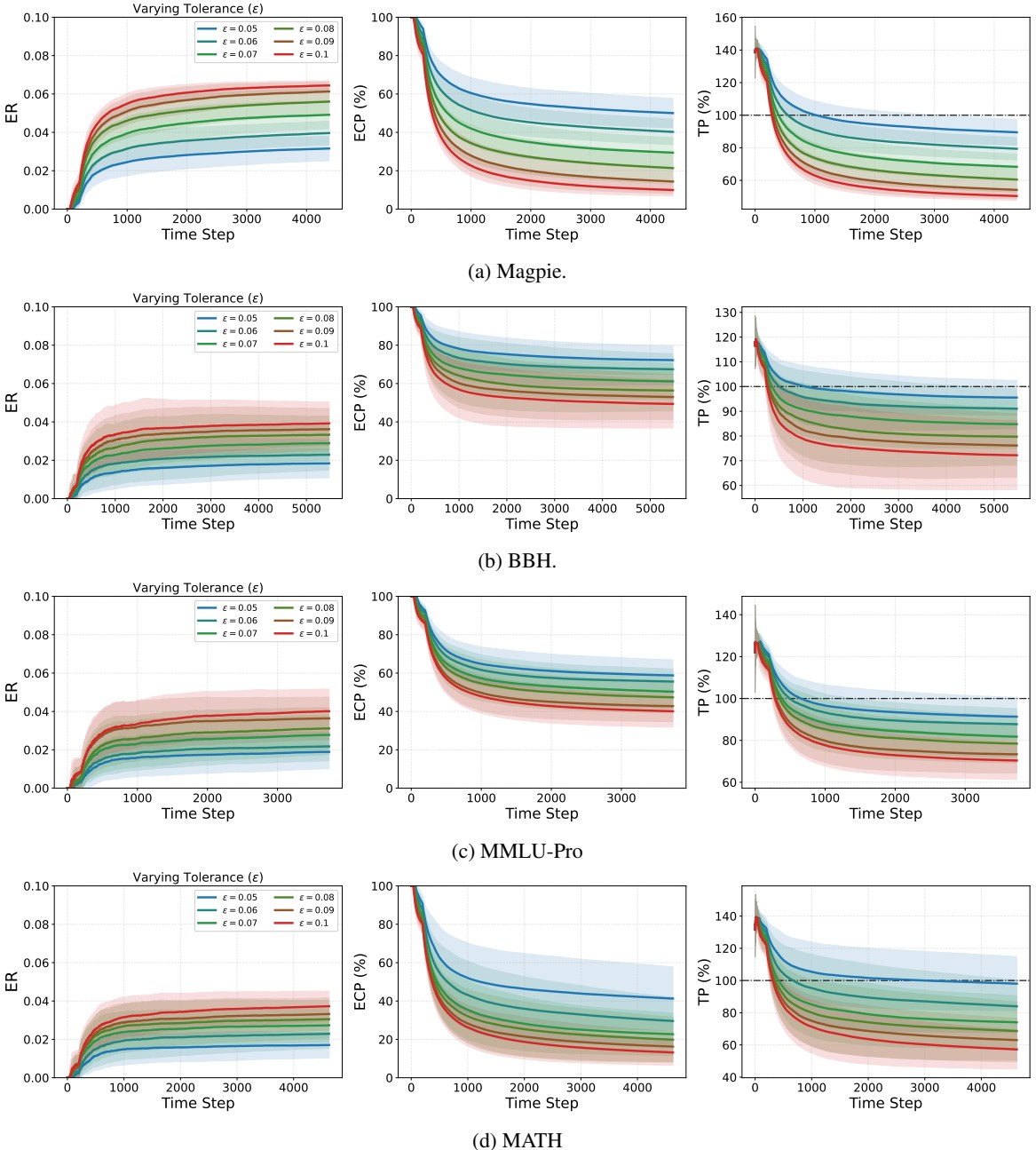

*Figure 5.* **Performance of B-PAC reasoning under different tolerance levels**. We consider $\epsilon \in \{0.05, 0.06, 0.07, 0.08, 0.09, 0.10\}$ on four benchmarks including Magpie, BBH, MMLU-Pro, and MATH. B-PAC reasoning ensures anytime-valid performance loss control across all settings, and achieves higher efficiency as $\epsilon$ increases.

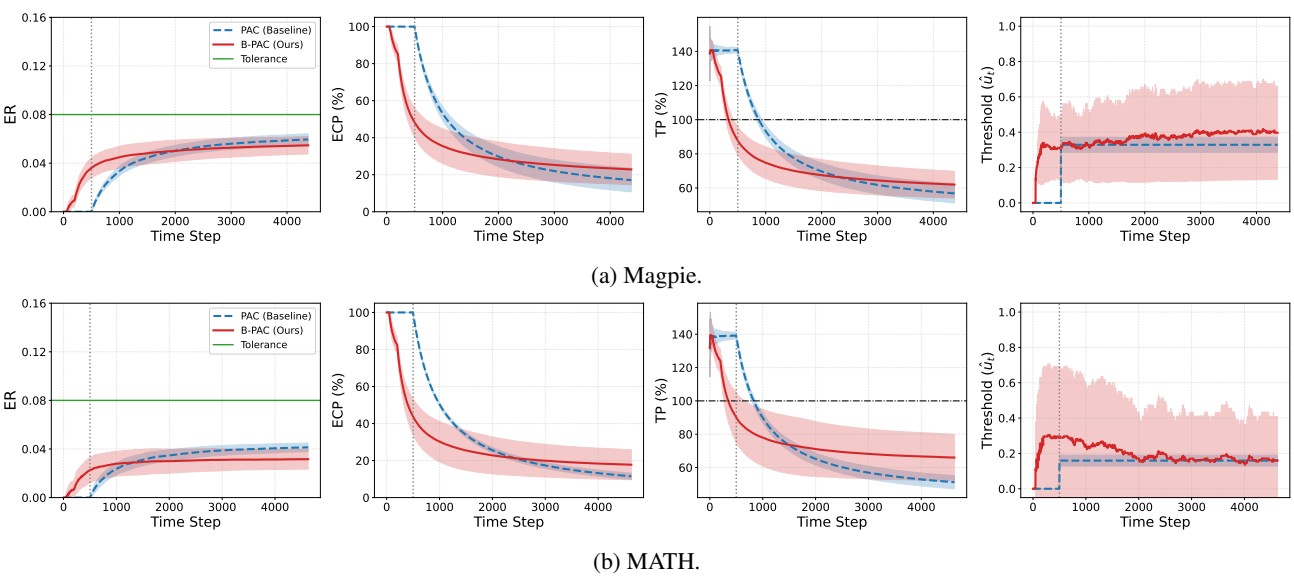

Figure 6. **Comparison with PAC reasoning on Magpie and MATH**. The results indicate that even under stationary data, B-PAC reasoning can approach or even surpass the efficiency of PAC by continuously updating the threshold.

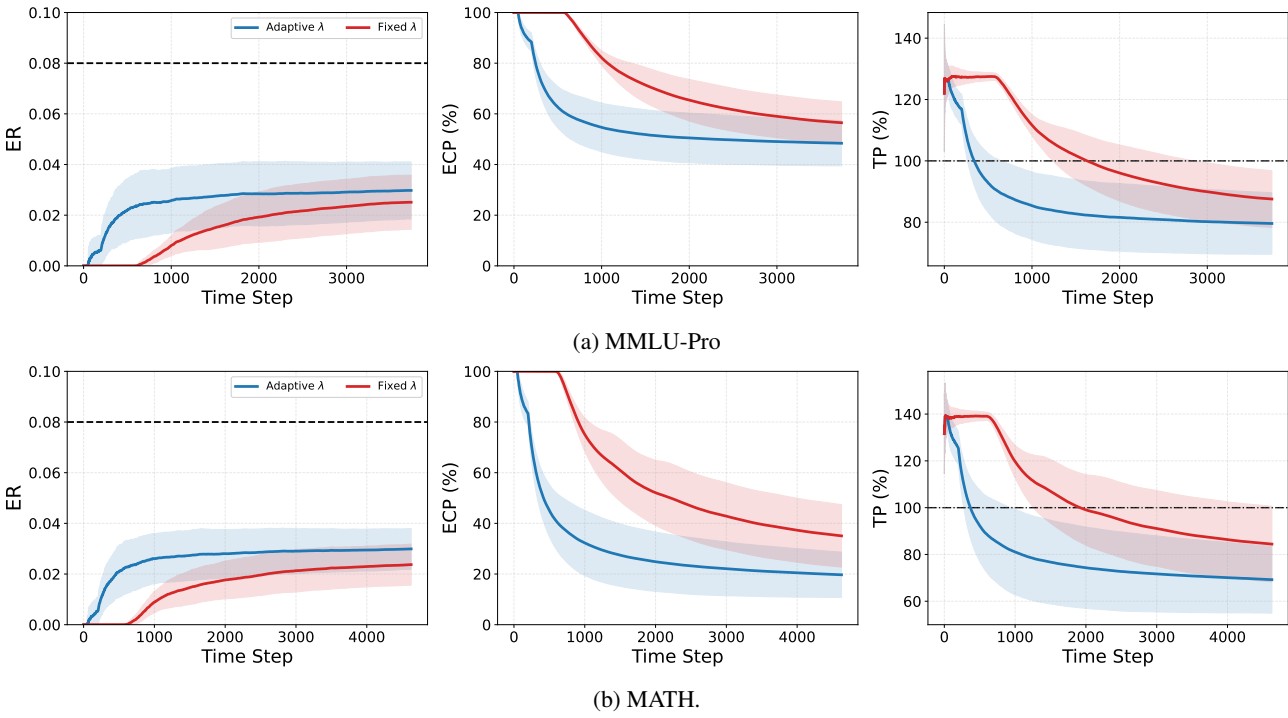

Figure 7. **Ablation study on the betting strategy $\lambda_t$ on MMLU-Pro and MATH.** The blue curve represents the adaptive betting strategy given by (8). The red curve represents the fixed betting strategy with $\lambda_t \equiv 0.05$. Results demonstrate that the adaptive betting strategy is more efficient compared to the fixed betting strategy.

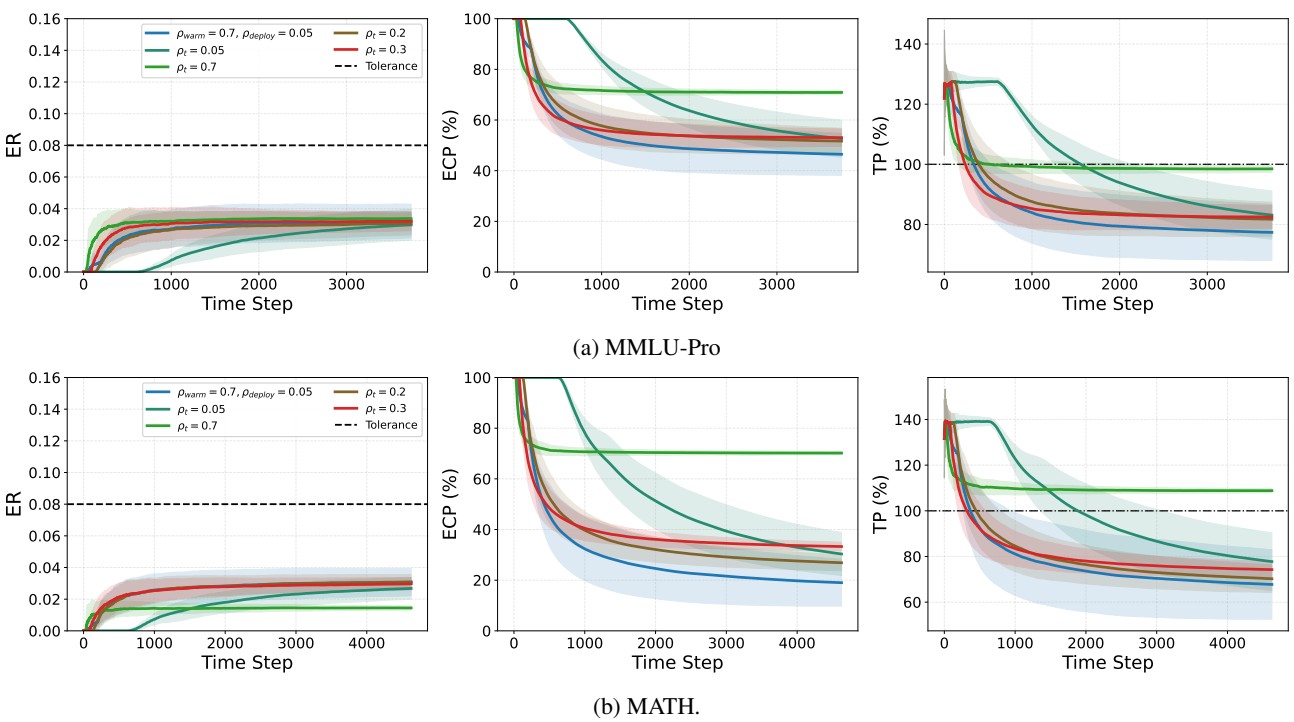

(a) MMLU-Pro

(b) MATH.

*Figure 8.* **Ablation study on the exploration probability** $\rho_t$. We compare the performance of our proposed two-stage exploration strategy (blue) against various fixed exploration probabilities $\rho_t \in \{0.05, 0.2, 0.3, 0.7\}$ on MMLU-Pro and MATH. Results indicate that the two-stage exploration strategy is more efficient compared to the fixed exploration strategy.

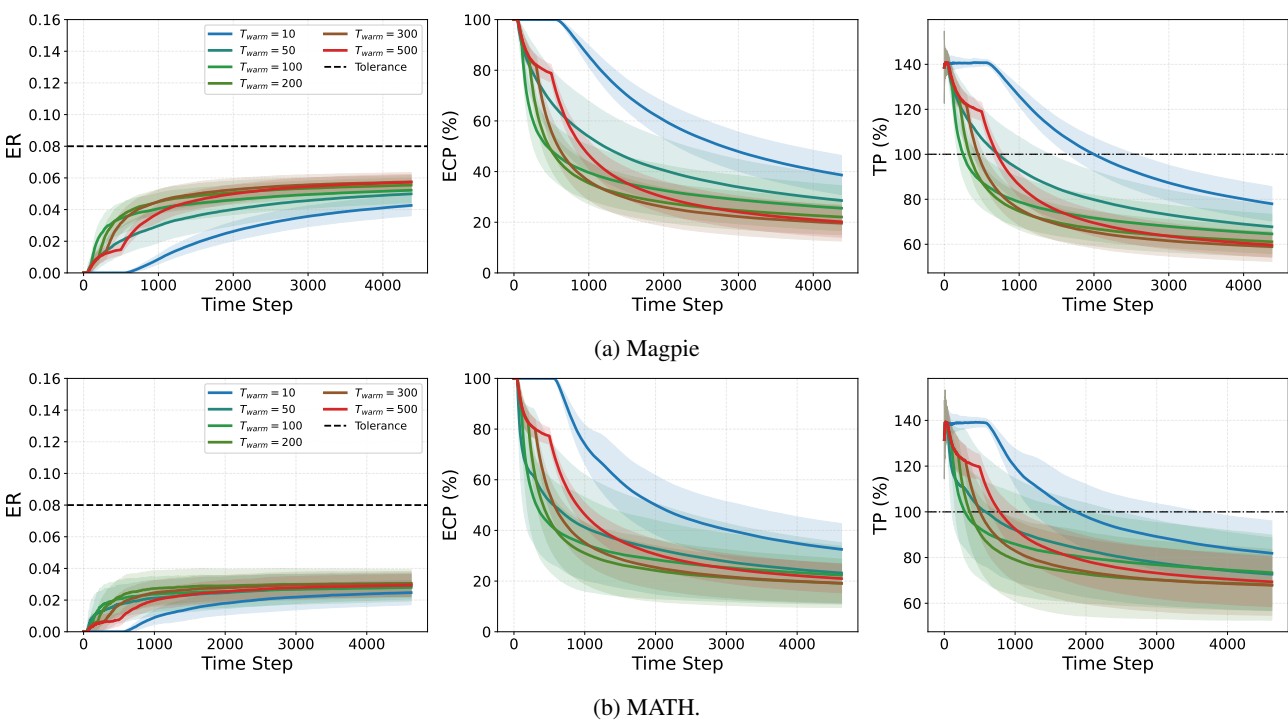

(a) Magpie

(b) MATH.

*Figure 9.* **Sensitivity to Warm-up Duration** $T_{\text{warm}}$. We compare the performance of B-PAC reasoning for $T_{\text{warm}} \in \{10, 50, 100, 200, 300, 500\}$ on Magpie and MATH. Results demonstrate the robustness of medium-sized $T_{\text{warm}}$.

