# OpenReview forum: "Anytime Safe PAC Efficient Reasoning"
_ICML.cc/2026/Conference — ICML 2026 regular_

### Official Review · Reviewer_mT6G · 2026-03-11

**Soundness:** 2
**Presentation:** 2
**Significance:** 3
**Originality:** 3
**Overall Recommendation:** 4
**Confidence:** 2

**Summary:**

This paper studies the problem of improving the efficiency of large reasoning models (LRMs) while controlling performance degradation in online settings. Existing selective reasoning approaches typically route easy queries to cheaper models to reduce computational cost, but they often lack rigorous guarantees on performance loss, especially when feedback is partial and data distributions may change over time. This paper proposes betting probably approximately correct reasoning, an online framework that provides anytime-valid safety guarantees.

**Compliance With Llm Reviewing Policy:**

Affirmed.

**Key Questions For Authors:**

No more.

**Limitations:**

Yes.

**Strengths And Weaknesses:**

Strengths：
1. Unlike many existing methods that are only valid under a fixed sample size or in offline evaluation settings, the proposed approach constructs statistical evidence through test supermartingales, enabling reliable risk control even when the system operates continuously in an online environment.

2. The method is able to learn under partial feedback, where the true error cannot be fully observed. It maintains unbiased estimation and theoretical safety guarantees, making the approach more suitable for practical deployment in real-world online systems.


Weaknesses：
1. The theoretical guarantees of the proposed method rely on several statistical assumptions, such as the use of the IPS estimator and certain distributional or independence conditions. In practical online systems, these assumptions may not always hold. For example, the data distribution may exhibit strong non-stationarity.

2. The evaluation is conducted on a limited set of models and datasets, and lacks experiments in more complex or realistic online system scenarios.

3. The B-PAC framework requires continuously maintaining test supermartingale statistics and updating them online. While this design is theoretically sound, it may introduce practical challenges in real-world deployment, including: (1) additional computational and system complexity, and (2) potential difficulties in parameter or threshold design and implementation.

---

> ### Author Rebuttal · Authors · 2026-03-31
>
> We sincerely thank the reviewer for the helpful comments and for recognizing `reliable risk control `, `learn under partial feedback`, `more suitable for practical deployment`. Below we address your concerns point by point.
>
> **1. Statistical assumptions:**
>
> Thank for your valuable comments. We clarify that B-PAC is **model-free** and applies to **non-stationary** settings, as reflected in Theorems 4.2 and 5.1 and in Figures 1 and 2. **No specific distributional assumption is required**, and the unbiased **IPS estimator is construted, not assumed**. To make this clearer, we list the assumptions below and explain their roles.
>
> - **(Only i.i.d. assumption)** We first establish B-PAC for i.i.d. streaming data in Section 3. The goal is to present the main idea clearly and leverage the fixed-sequence testing to obtain a highly efficient model (Remark 2.2). Under the i.i.d. assumption, the deployment risk $R(u)$ is equals to $(1-\rho_{deploy})\mathbb{E}[l_t\mathbb{I}\{U_t<u\}]$, a constant unrelated to $t$. This allows us to construct a test supermartingale and leverage Ville’s inequality and fixed-sequence testing. Finally, by using $R_t(u)\leq R(u)$, we complete the proof of Theorem 4.2. See Page 13-15 for details.
>
> - **(Beyond i.i.d.)** In section 5, we drop the i.i.d. assumption and extend B-PAC to non-stationary settings. The main challenge is that the underlying risk $r_t=(1-\rho_t)\mathbb{E}[l_t\mathbb{I}\{U_t<u\}|\mathcal{F}_{t-1}]$ can vary with $t$, making the proof of Theorem 4.2 infeasible. We address this by modifying the threshold-selection criterion to Equation (12) and leveraging the tool of mixture martingales to establish safety. Figures 1 and 2 further demonstrate that B-PAC remains safe and efficient under non-stationary data.
>
> **2. Evaluations on more LLMs and dataset:**
>
> Thanks for your helpful comments.
> - **(More models)** We add new results on **Deepseek-r1-distill-qwen-14b** and **Deepseek-r1-distill-llama-8b**; please refer to our response to **the first question of Reviewer g7ZL (the second Table)** for details.
>
> - **(More complex settings)** Compared with existing work on selective reasoning, our evaluated benchmarks: MATH, MMLU-Pro, BBH, and Magpie, are already comprehensive, covering mathematical problem solving, multi-task knowledge, symbolic reasoning, and open-ended instruction following. We also evaluated non-stationary online settings, where the type and difficulty of query can vary substantially over time; see **Figures 1 and 2**. To further demonstrate applicability in complex settings, **we add new experments results** on a mixed dataset combining MATH, MMLU-Pro, and BBH, with settings identical to the original manuscript. Note that $\epsilon=0.08$.
>
> |Method|ER|ECP(%)|TP(%)|
> |-|-:|-:|-:|
> |PAC (Baseline)|0.0945 ± 0.01|28.09 ± 4.21|58.40 ± 5.31|
> |B-PAC (Ours)|0.0582 ± 0.01|49.46 ± 6.35|84.30 ± 7.00|
>
> The results show that B-PAC is safe (ER $<0.08$) and efficient. This experiment also demonstrates **the usability of B-PAC in online scenarios where task type and difficulty can change drastically.**
>
> **3.Computational and system complexity, parameter/threshold design and implementation:**
>
> We sincerely thank you for your considerations. These issues are discussed in **Appendix A.3 and D**. Here, we provide a point-by-point response.
>
> **(1) Computational and system complexity are negligible**
>
> - **Compute Overhead:** Briefly, the total systematic computational cost of B-PAC for **1000 requests** is only **0.046 seconds**. This is because updating $\hat{u}_t$ requires no model retraining, gradient updates, or heavy matrix multiplications. By equation (7), we only maintain $K_t(u)$ over a discretized threshold grid, which just involves computing the IPS estimator and performing scalar multiplications per step.
>
> - **System complexity:** Systemically, B-PAC acts only as a lightweight routing layer (a "state tracker"). It requires storing only a **1D float array** of size $|\mathcal{U}|$ in memory (or a KV store such as Redis) to persist the wealth state across streaming API requests.
>
> **(2) Principled parameters, adaptive threshold, and easy implementation:**
>
> - **Hyperparameters:** B-PAC only involves hyperparameters $\rho_{warm}, \rho_{deploy}, T_{warm}$, and their roles are interpretable. We also provide robust choices and ablation experiments. Please refer to our response to the **first question of Reviewer FUpJ** for details.
>
> - **Adaptive threshold:** The candidate set $\mathcal{U}$ can be implemented as a uniform grid, e.g., np.linspace(0, 1, 100). B-PAC then automatically determines $\hat{u}_t$.
>
> - **Easy implementation:** As noted above, B-PAC is fully automated, with negligible additional memory and computational cost. We also discuss more advanced engineering issues, including **asynchronous state management and scalability via distributed betting**; see Appendix A.3 for details.
>
> In the final version, we will present these points more clear in the main context.

---

> > ### Author Rebuttal · Reviewer_mT6G · 2026-04-03
> >
> > Thank you for your effort, I will maintain my positive initial score.

---

> > > ### Author Response · Authors · 2026-04-03
> > >
> > > Thank you for your careful follow-up. We are pleased that our response fully addressed your concerns. Your feedback is highly valuable in improving the quality of this work. Once again, we appreciate your time and positive feedback.

---

### Official Review · Reviewer_FUpJ · 2026-03-12

**Soundness:** 3
**Presentation:** 3
**Significance:** 4
**Originality:** 3
**Overall Recommendation:** 5
**Confidence:** 4

**Summary:**

This paper identifies inefficiencies when Large Reasoning Models generate unnecessarily long chains of thought for simple queries. One solution involves routing these simpler queries to a cheaper, faster model, but doing so risks accidentally routing complex queries to the cheap model, which can produce confidently wrong answers with no way of knowing how often it's happening. The authors solve this by framing the routing decision as a betting game, where statistical evidence accumulates over time to determine how aggressively the system can rely on the cheap model. A running "wealth" process tracks whether a given routing threshold is safe. If the cheap model keeps performing acceptably, wealth grows and the system routes more queries its way; if errors creep up, wealth shrinks and the system pulls back. This method gives a hard guarantee that the cheap model's error rate stays below a limit a user defines in advance at every point in time. Experiments show the method works across a range of benchmarks, but the authors themselves note it only handles binary routing between two models, and its efficiency depends on the quality of the uncertainty score used to classify queries in the first place.

**Compliance With Llm Reviewing Policy:**

Affirmed.

**Ethical Review Concerns:**

False alarm: no ethical issues

**Final Justification:**

My initial concerns centered on hyperparameter sensitivity, the binary routing constraint, and the stability of safety guarantees under noisy LLM-as-a-judge loss functions. The rebuttal addressed all three convincingly. On hyperparameters, the authors demonstrated that safety is invariant to these choices and provided both theoretical justification and empirically robust defaults. On multiple models, they sketched a concrete extension path and identified the main technical challenge, which is reassuring even if the work is left to future iterations. On noisy loss, they showed formally that safety holds under additive noise, with noise affecting variance rather than the guarantee itself. Taking all of this into account, I am upgrading my recommendation from a 4 to a 5. The remaining limitations (binary routing, uncertainty score dependence) are acknowledged, and neither undermines the core contribution.

**Key Questions For Authors:**

1. The two-stage exploration strategy requires setting ρ_warm, ρ_deploy, and T_warm before deployment. How sensitive are the safety and efficiency results to these choices, and is there a principled way to set them without domain knowledge?
2. The method is restricted to two models. Is there a fundamental barrier to extending the betting framework to multiple models, or is this purely an engineering gap?
3. For open-ended benchmarks, the paper uses an LLM-as-a-judge loss. How stable is the safety guarantee when the loss function itself is noisy or inconsistent? If the judge disagrees with itself across runs, the IPS estimator and wealth process could behave unpredictably.

**Limitations:**

yes

**Strengths And Weaknesses:**

The theoretical foundation is good, specifically how the statistical wealth process behaves exactly as the theory requires (shrinking when a threshold is risky and growing when it isn't). Moreover, the IPS correction is well-motivated, and the authors are honest about where the method falls short. The presentation builds logically from problem to method to theory to experiments. Controlling error rates in a live, shifting deployment is a legitimate infrastructure problem, thus the significance of this work is strong, and the extension to non-stationary data has applications for real-world settings. Treating the routing threshold selection as a betting game is a creative reinterpretation of the problem that comes with rigorous guarantees. Although real deployments often involve more than two models, the framework is general enough to have significant impact for online query routing.

---

> ### Author Rebuttal · Authors · 2026-03-31
>
> We sincerely thank the reviewer for the helpful comments and for recognizing `theoretical foundation is good`, ` well-motivated`, `presentation builds logically`, `the significance of this work is strong`, ` a creative reinterpretation ... with rigorous guarantees`, and `have significant impact for online query routing`. Our point-by-point responses are as follows.
>
> **1. Hyperparameters:**
>
> Thanks for this important question. The impacts and ablation studies of these parameters are provided in **Section 3.3** and **Appendix D.4 (Figures 8, 9)**, respectively. Here, we summarize the key points.
> - **(Safety is insensitive to hyperparameters)** For any $(\rho_{warm},\rho_{deploy})\in(0,1)\times(0,1)$ with $\rho_{warm}\geq \rho_{deploy}$ and any $T_{warm}$, the safety of B-PAC always holds by Theorem 4.2 and 5.1. Thus, tuning affects efficiency, not safety.
> - **(Efficiency tradeoff)** A large $\rho_t$ permits more aggressive betting via $\lambda_t$ and $D_t$, so the wealth $K_t(u)$ grows faster for safe $u$, helping identify a tight safe threshold early. After warm-up, a smaller $\rho_t$ is preferable since the expert-call probability is at least $\rho_t$. This motivates $\rho_{warm}\geq \rho_{deploy}$ with a suitable $T_{warm}$. Empirically, $\rho_{warm}=0.7$, $\rho_{\text{deploy}}=0.05$, and $T_{warm}=200$ are robust across all our experiments.
> - **(Principled insight)** Optimal parameter selection is generally impossible due to the black-box nature of LLMs (e.g., the distribution of $\tilde{f}(X)$ is unknown). Nonetheless, we give an insight for i.i.d. data: **the asymptotic efficiency is dominated by $\rho_{deploy}$.** A concise analysis is as follows and the notations follow the paper. Let $q_0$ be the maximum threshold for controlling deployment risk. For safe $u$, martingale-difference SLLN gives
> $$\frac{1}{t}\sum_{i=1}^t(\epsilon-Z_i(u))\overset{a.s.}{\to}\epsilon-R(u).$$
> Through simple calculations, $\hat{u}\_t \to q_0$ almost surely, implying $\mathbb{E}[\xi_{t}] \to \rho_{deploy} \mathbb{P}(U_t < q_0) + \mathbb{P}(U_t>q_0)$ as $t \to \infty$. Therefore, $\rho_{deploy}$ should be small for long-time deployment. Considering practical non-stationary, we empirically recommend $\rho_{deploy}=0.05$.
>
> Overall, we believe the above analysis is sufficient to guide practical deployment of B-PAC.
>
> **2. Multiple models:**
>
> Thanks for this insightful question. Extending B-PAC to multiple models presents **no fundamental barrier, but requires careful design**. Let $f^{(k)}$, $k=1,\dots,K$, denote models with increasing capability and cost. For simplicity, let $K=3$. We aim to build a composite model $\hat{f}_t(X_t)\in\{f^{(k)}(X_t):k=1,2,3\}$ that controls loss w.r.t. $f^{(3)}(X_t)$ with high probability.
>
> Let $\rho_t$ be the minimum exploration probability and $I_t=\text{Bernoulli}(\rho_t)$. If $I_t=1$, we output $f^{(3)}(X_t)$ and call the cheaper models to compute loss. If $I_t=0$, we use the threshold vector $(\hat{u}\_{t-1}^{(1)},\hat{u}\_{t-1}^{(2)})$ to split uncertainty space. We first compute $f^{(1)}(X_t)$ and its uncertainty score $U_t^{(1)}$. If $U_t^{(1)}<\hat{u}\_{t-1}^{(1)}$, we stop; otherwise, we compute $f^{(2)}(X_t)$ and check whether $U_t^{(2)}<\hat{u}\_{t-1}^{(2)}$. If so, we stop; otherwise, we output $f^{(3)}(X_t)$. For a vector $\mathbf{u}$, we denote the resulting model by $f_{\mathbf{u}}$, and construct an IPS estimator by $Z_t(\mathbf{u})=\frac{I_t}{\rho_t}l(f_{\mathbf{u}}(X_t),f^{(3)}(X_t))$. Following B-PAC, we define the wealth $K_{t}(\mathbf{u})=K_{t-1}(\mathbf{u})(1+\lambda_t(\mathbf{u})(\epsilon-Z_t(\mathbf{u})))$. The admissible threshold is defined similarly as B-PAC. The main challenge lies in efficient optimization over a two-dimensional space. Hence we leave this extension to future work.
>
> Overall, this analysis suggests that extending B-PAC to multiple models is promising, though further work is needed to improve efficiency.
>
> **3. Validity for random loss:**
>
> We thank you for raising this practical concern. **Briefly: safety remains valid; noise mainly affects efficiency.**
>
> Practically, we recommend two strategies: using Temperature=0 for the LLM judge to remove randomness, or averaging over multiple judges to stabilize the loss.
>
> Theoretically, let $\tilde{l}(\hat{f}_t(X),f(X); Q_t)$ be the random loss, where noises $(Q_t)\_{t\geq 1}$ are i.i.d. and independent of $(X_t)\_{t\geq1}$. By the same proof, we have $\mathbb{P}(\forall t\geq1:V_t(\hat{f}_t)\leq \epsilon) \geq 1-\alpha,$ where $V\_t(\hat{f}\_t) =\mathbb{E}\_{X\sim P\_X,Q\_t\sim Q}[\tilde{l}(\hat{f}\_t(X),f(X);Q_t)].$ That is, safety still holds, but for $V_t(\hat{f}_t)$ rather than $R_t(\hat{f}_t)$ (Definition 1). In the common additive-noise model $\tilde{l}(\hat{f}_t(X),f(X);Q_t)=l(\hat{f}_t(X),f(X))+Q_t$ with mean-zero noise, we have $V_t(\hat{f}_t)=R_t(\hat{f}_t)$. Therefore, safety remains; noise only introduces extra variance and may reduce efficiency.
>
> We will explicitly discuss this in the final version.

---

> > ### Author Rebuttal · Reviewer_FUpJ · 2026-04-02
> >
> > The authors directly addressed all three questions. On hyperparameters, they showed safety holds regardless of choices and provided empirical defaults with theoretical justification. On multiple models, they sketched a concrete extension and identified the main technical challenge. On noisy loss functions, they showed safety remains valid under additive noise. I have updated my score to reflect this.

---

> > > ### Author Response · Authors · 2026-04-03
> > >
> > > We sincerely thank you for reviewing our response and increasing the score. We are delighted that our response addressed your concerns. Your feedback is highly valuable in improving the quality of this work.

---

### Official Review · Reviewer_g7ZL · 2026-03-16

**Soundness:** 4
**Presentation:** 4
**Significance:** 3
**Originality:** 3
**Overall Recommendation:** 5
**Confidence:** 4

**Summary:**

Large reasoning models (LRM) provide better answers, especially on complex tasks, than non-reasoning models at a higher token cost. This has inspired a number of methods in recent years to switch between reasoning and non-reasoning models depending on the model uncertainty of the cheaper non-reasoning model. However, these methods do not control the performance loss brought about by this computational gain. While there has been some work on risk control and PAC reasoning, it requires a separate calibration set which is not ideal for online inference.

This paper proposes betting PAC (B-PAC) for safety in reasoning by controlling the performance of switching reasoning models online. The method assumes two models f_tilde and f, where f_tilde is a non-reasoning approximation to the reasoning model f, which for the purposes of the analysis is taken as true. The decision of whether to route to f is sampled from a stochastic policy that depends on whether the uncertainty score for f_tilde exceeds a dynamic threshold, with some exploration rate if the uncertainty is low. The threshold is selected by constructing an IPS estimator of the loss of using f_tilde then constructing a non-negative supermartingale to determine the most efficient model. This approach is discussed with a betting explanation where the supermartingale process is interpreted as the accumulated capital of a gambler testing the null hypothesis that the threshold is unsafe. B-PAC is evaluated on several reasoning datasets under two versions of QWEN (reasoning and instruct) against several appropriate baselines showing safe and efficient outcomes for B-PAC.

**Compliance With Llm Reviewing Policy:**

Affirmed.

**Final Justification:**

I recommend acceptance because the paper presents a well-motivated, clearly written, and theoretically sound framework for anytime-safe, efficient routing between reasoning and non-reasoning models, with rigorous guarantees that hold even under partial feedback and non-stationarity. The rebuttal convincingly addressed my main concern about generality by adding experiments on additional models and uncertainty scores and clarifying broader applicability, which reinforced rather than changed my positive assessment.

**Key Questions For Authors:**

1. It appears that none of the analysis is specific to reasoning per se. Knowing that reasoning models are more expensive because they cost more in test time inference with more tokens compared with non-reasoning model, is there a way to make a tighter bound?
2. Since the method seems quite general, is B-PAC applicable to other settings where is a cheaper and more expensive model? Could the more expensive "model" be human?
3. Where did the IPS+Hoeff line go in Figure 3?

**Limitations:**

Yes, they acknowledge that B-PAC does not work for > 2 models and that it depends on the quality of confidence scores, making further explorations on confidence in empirical results important here.

**Strengths And Weaknesses:**

Strengths:
+ strong motivation and background: we want to control the safety of relying on non-reasoning models when uncertainty is low
+ excellent presentation with clear introduction and discussion of the technical aspects of the work
+ the approach of treating prompts potentially routed to a reasoning model as a partial feedback setting makes sense and is very interesting
+ theory-backed approaches to selecting the nuisance hyperparameters $\lambda$ and $\rho$

Weaknesses:
+ while several datasets were used in experiments only one base model essentially was used (Qwen3-4B) and only one logits-based uncertainty score, so it's not clear how the method works on other models and uncertainties

---

> ### Author Rebuttal · Authors · 2026-03-31
>
> We sincerely thank the reviewer for the insightful comments and for recognizing `strong motivation and background`, `excellent presentation`, `the approach ... makes sense and is very interesting`, `theory-backed approaches`. Below, we address your concerns point by point.
>
> **1. Additional experiments**
>
> We thank you for this helpful suggestion. Theoretically, B-PAC reasoning is **model-agnostic** (i.e., it can be applied to various LLMs) and puts **no specific restrictions on uncertainty score** (see Remark 3.1). To further strengthen this claim, we **add new experiments** as follows. Basic settings are identical to those in the original manuscript. All values are reported as mean ± standard deviation, and we note that $\epsilon=0.08$.
>
> - **Other uncertainty scores:**  Results for uncertainty scores including **Perplexity (PPL)** and **Entropy**.
>
> |Dataset|Uncertainty Score|ER|ECP(%)|TP(%)|
> |-|-|-:|-:|-:|
> |Magpie| PPL |0.0551 ± 0.007|22.76 ± 7.57|61.69 ± 7.22|
> |Magpie|Entropy|0.0552 ± 0.006|23.12 ± 6.51| 61.84 ± 6.18|
> |MMLU-Pro|PPL|0.0319 ± 0.011|43.24 ± 8.19|75.12 ± 9.26|
> |MMLU-Pro|Entropy|0.0316 ± 0.012|43.08 ± 8.66|74.59 ± 9.83|
>
> - **Other models:** Results for **Deepseek-r1-distill-qwen-14b** and **Deepseek-r1-distill-llama-8b** (thinking model).
>
> |Dataset|Model|ER| ECP(%)| TP(%)|
> |-|-|-:|-:|-:|
> |MMLU-Pro|Deepseek-r1-distill-qwen-14b|0.0310+-0.009| 42.55+-6.77| 85.41+-7.55|
> |MMLU-Pro|Deepseek-r1-distill-llama-8b| 0.0303±0.011| 35.22±7.68| 68.53±7.09|
>
> Overall, B-PAC remains safe (ER<0.08) and efficient (small ECP and TP) across different models and uncertainty scores. We will add the above results into the revised manuscript.
>
> **2. Tighter bound**
>
> We thank you for pointing out the generality of B-PAC, namely that it is not limited to reasoning models. We intentionally designed B-PAC in a model-agnostic manner, since existing LLMs are highly black-box and **any model-specific methods may be restricted or risky for LLM applications.**
>
> Your suggestion is very insightful: if we can incorporate inherent characteristics of reasoning models, the efficiency of B-PAC might be further improved. Unfortunately, **a tighter bound is unlikely to be derived without further statistical assumptions.** **First**, to the best of our knowledge, game-theoretic statistics, key techniques for proving the safety of B-PAC, already represent the state-of-the-art technique for anytime-valid inference [1]. **Second**, the realized risk and the efficiency depend on the quality of $U_t$ for $\tilde{f}(X_t)$, rather than on the number of tokens or the test-time cost of the thinking model $f$. **Third**, as demonstrated by Figure 2 (left panel), the achieved risk is already sufficiently close to the target level.
>
> However, this suggestion points to a promising direction: **cost-aware routing**. Taking the i.i.d setting as an example, the current goal of B-PAC is to $\min \mathbb{E}[\mathbb{I}\{\xi_t =1\}]$ under the premise of safety. Since reasoning models may generate significantly more tokens for some queries, the penalty for routing a query to $f$ should vary across inputs. This motivates the goal of $\min \mathbb{E}[C(X_t)\mathbb{I}\{\xi_t=1\}]$, where $C(X_t)$ denotes the cost of the thinking model output $f(X_t)$. In practice, one may obtain $\hat{C}(X_t)$ by using prediction models, and then threshold updating relies on both $U_t$ and $\hat{C}(X_t)$. This allows the system to route moderately easy but highly token-intensive queries more aggressively to the non-thinking model, thereby achieving greater token savings under the same risk constraint. We will explicitly indicate in the revised manuscript the extension of B-PAC in this direction.
>
>
> **3. Extension to settings beyond reasoning**
>
> We thank the reviewer for this forward-looking question. It is absolutely right that the theoretical foundation of B-PAC is highly general and **directly applies to any routing system with a cost-accuracy trade-off** (i.e., a cheaper model $\tilde{f}$ and a more expensive model $f$).
>
> Furthermore, the promising idea of treating the "expensive model" as a human **perfectly maps B-PAC onto the Human-in-the-Loop and Learning to Defer paradigms**. In high-stakes applications (e.g., medical diagnosis) and reliable automatic labeling, B-PAC can dynamically optimize the deferral threshold to minimize expensive human labor while providing rigorous loss control. Since this observation suggests that B-PAC can have a broader impact on several important fields, we will explicitly discuss these applications in the revised manuscript.
>
> **4. Figure presentation**
>
> We thank you for this question. The IPS+Hoeff curve is plotted in Figure 3 but may be difficult to distinguish since it lies near the plot boundary (e.g., ECP close to 100%). We will make this clearer in the final version.
>
> ### Reference
> [1] Ramdas et al. (2023). Game-theoretic statistics and safe anytime-valid inference. Statistical Science, 38(4):576–601.

---

> > ### Author Rebuttal · Reviewer_g7ZL · 2026-04-02
> >
> > My questions have been addressed. I maintain my score.

---

> > > ### Author Response · Authors · 2026-04-03
> > >
> > > Thank you for your positive feedback and for carefully reviewing our response. We are pleased that our response addressed your concerns, which also improves the quality of this work. Once again, we appreciate your positive and valuable feedback.

---

### Decision · Program_Chairs · 2026-04-30

**Decision:**

Accept (regular)

**Comment:**

This paper considers the challenge of balancing efficiency and safety in Large Reasoning Models (LRMs) by proposing B-PAC, a model-agnostic framework that uses a betting-based mechanism to dynamically route queries between thinking and non-thinking models. While the proposed method provides a principled way to maintain anytime-valid performance loss control in online, non-stationary settings, the reviewers initially raised several concerns. Specifically, they questioned the sensitivity of hyperparameters, the stability of the safety guarantees under noisy evaluation (such as LLM-as-a-judge), and the generalizability across different models and uncertainty scores.

Since the authors addressed these points thoroughly in their rebuttal by providing additional experiments on DeepSeek models and theoretically proving that safety guarantees hold regardless of hyperparameter tuning, the reviewers reached a positive consensus that their concerns were fully resolved.

Given the technical soundness and the strong empirical results, the recommendation is to accept.